# Difluoroester solvent toward fast-rate anion-intercalation lithium metal batteries under extreme conditions

Yao Wang[1,3], Shuyu Dong[2,3], Yifu Gao[1], Pui-Kit Lee[2], Yao Tian[1], Yuefeng Meng[1], Xia Hu[1], Xin Zhao[1], Baohua Li [1], Dong Zhou [1] ✉ & Feiyu Kang[1] ✉

Anion-intercalation lithium metal batteries (AILMBs) are appealing due to their low cost and fast intercalation/de-intercalation kinetics of graphite cathodes. However, the safety and cyclability of existing AILMBs are constrained by the scarcity of compatible electrolytes. Herein, we showcase that a difluoroester can be applied as electrolyte solvent to realize high-performance AILMBs, which not only endows high oxidation resistance, but also efficiently tunes the solvation shell to enable highly reversible and kinetically fast cathode reaction beyond the trifluoro counterpart. The difluoroester-based electrolyte demonstrates nonflammability, high ionic conductivity, and electrochemical stability, along with excellent electrode compatibility. The Li||graphite AILMBs reach a high durability of 10000 cycles with only a 0.00128% capacity loss per cycle under fast-cycling of $1\,A\,g^{-1}$, and retain ~63% of room-temperature capacity when discharging at −65 °C, meanwhile supply stable power output under deformation and overcharge conditions. The electrolyte design paves a promising path toward fast-rate, low-temperature, durable, and safe AILMBs.

Facing formidable environmental challenge, substantial progress in battery technology is paramount for enabling a transformative shift in energy paradigms, ultimately aiming for a society with reduced carbon footprints. As a promising candidate for next-generation battery system that offers higher energy density, lithium (Li) metal batteries (LMBs) are highly pursued owing to the unparalleled theoretical specific capacity ($3860\,mAh\,g^{-1}$) and the lowest redox potential (−3.04 V vs. standard hydrogen electrode) of Li metal anodes[1–3]. However, the practical implementation of LMBs has been plagued by the prevailing transition metal oxide-based cathodes, which not only suffer from high costs and raise concerns of an economically and geopolitically constrained supply, but also exhibit sluggish intercalation/de-intercalation kinetics and limited lifespan arising from cathode structure deterioration (e.g., transition metal dissolution, gas evolution)[4]. The later issues will be further exacerbated under fast-rate and low-temperature operating conditions[2]. Therefore, it is essential to eliminate transition

metal elements from cathode materials and thus, re-design battery chemistry for the developments of LMBs. In this regard, anion-intercalation LMBs (AILMB) have emerged by replacing the Li+-hosting transition-metal oxide cathodes with cost-efficient graphitic carbons as anion hosts[5,6]. In contrast to the "rocking-chair" mechanism in traditional LMBs based on transition-metal oxide cathodes, the simultaneous redox process on both the anion-intercalation cathode and cation-plating anode in AILMBs potentially weakens the de-solvation barrier, endowing this battery system with fast-rate and low-temperature characteristics[7]. However, the high operating voltage (5 V-class vs. Li/Li+) of the anion-intercalation chemistry at graphite cathodes causes irreversible side reactions with conventional electrolytes, leading to the formation of high-resistance cathode|electrolyte interphase (CEI) on the cathode surface which seriously inhibits anion insertion[8–10]. Moreover, the substantial and repeated cathode volume change (>130%) and the co-intercalation of solvent molecules trigger

[1]Tsinghua Shenzhen International Graduate School, Tsinghua University, Shenzhen 518055, China. [2]School of Energy and Environment, City University of Hong Kong, Hong Kong SAR 999077, China. [3]These authors contributed equally: Yao Wang, Shuyu Dong. ✉e-mail: zhou.d@sz.tsinghua.edu.cn; fykang@tsinghua.edu.cn

an exfoliation of graphite layers during cycling, significantly deteriorating the graphite cathode structure[11]. As for the Li metal anode, the fragile solid electrolyte interphase (SEI) and uncontrollable dendrite growth during cycling results in low Coulombic efficiency (CE) and even catastrophic safety issues (e.g., internal short circuit)[3,12,13]. These drawbacks lead to the degraded cycle life and serious self-discharge in existing AILMBs[8,14].

Electrolyte engineering is promising for enhancing the electrode| electrolyte compatibility, thereby achieving durable AILMBs. Ether-based electrolytes have been widely employed in traditional LMBs due to their relatively low reactivity with Li metal[15,16]. Nevertheless, they are incompatible with anion-intercalation cathodes because of their intrinsic oxidative instability at high voltage (<4 V vs. Li$^+$/Li)[15,17]. In contrast, linear carbonates represented by ethyl methyl carbonate (EMC) endow a reversible anion intercalation/deintercalation on graphite cathode in AILMBs. However, the safety concerns originating from their high flammability, the insufficient stability of linear carbonates toward Li metal anode together with solvent co-intercalation on the cathode, remain to be solved[18,19]. Although applying high-concentration salt can alleviate above issues via reducing the proportion of free solvent molecules, the high salt cost along with high viscosity and poor electrode wettability of such concentrated electrolytes significantly restrict the application feasibility[20,21]. Recently, ester solvents have been widely utilized to enhance the low-temperature and fast-charging performance of Li-ion batteries benefiting from their low freezing points and low viscosities[22,23]. However, their poor compatibility with Li metal anode and limited oxidation stability (<4.7 V vs. Li$^+$/Li)[24] greatly block their application in AILMBs; to the best of our knowledge, the esters have not been reported as a solvent in AILMB system yet. Thereby, designing highly compatible electrolyte systems for safe, durable, fast-charging, and low-temperature AILMBs remains a significant challenge.

Herein, we systematically investigated a family of fluorinated esters as the solvent for the electrolytes of AILMBs. Our findings reveal that fluorination to ester molecules effectively enhances both the anti-oxidative property and stability with Li anode. Unexpectedly, as verified by computational modeling and experimental results, compared with the trifluoro (-CF$_3$) counterpart, the difluoroester (-CHF$_2$) effectively attenuates the anion-solvent interactions, thereby reducing corresponding anion de-solvation kinetic barrier and suppressing solvent co-intercalation into graphite cathodes. This balanced electrolyte design enables highly reversible and kinetically fast anion intercalation. The difluoro 2,2-difluroethyl acetate (DFEA)-based electrolyte demonstrates high ionic conductivity (7.2 mS cm$^{-1}$, 25 °C), remarkable electrochemical stability (up to 5.5 V vs. Li$^+$/Li), excellent compatibility with the Li metal anode and high safety without combustion concerns. Under fast-cycling condition of 1 A g$^{-1}$, the as-developed AILMB exhibits record-high durability of 10,000 cycles with a capacity retention of 88.0% (with negligible capacity fade of 0.00128% per cycle only), much beyond the reported LMBs based on LiFePO$_4$ (LFP) or LiNi$_{0.8}$Co$_{0.1}$Mn$_{0.1}$O$_2$ (NCM811) cathodes. More importantly, this AILMB remains ~63% of room-temperature (RT, 25 °C) capacity when discharging at −65 °C, and offers steady power output of pouch cells under abusive conditions (e.g., deformation and overcharge). This work represents a significant advancement in LMB performance and is particularly advantageous for applications in extreme conditions (e.g., fast-charging electric vehicles, aerospace and polar region devices).

## Results

### Screen of ester solvents and investigation on anion-intercalation

It is known that despite the merits including appropriate dielectric constant (6.02), low viscosity (0.45 mPa s), and low melting point (−84 °C)[22], ethyl acetate (EA) exhibits large overpotential and poor Li stability in LMBs originating from the high de-solvation energy and the inability to form a protective SEI, respectively[22]. Besides, as seen from inset of Fig. 1a, the 1.2 M LiPF$_6$ salt in EA electrolyte demonstrates high flammability with a self-extinguishing time (SET) of 110 s g$^{-1}$ due to the high volatility of EA solvent. More seriously, the EA-based electrolyte fails to support the anion-intercalation chemistry on the graphite cathode. During the initial charging process at 20 mA g$^{-1}$ (0.2 C) in an AIMIB, this electrolyte exhibits an abnormal charging profile, characterized by a large irreversible charging capacity and an inability to reach the cutoff voltage of 5.2 V, which can be attributed to the continuous electrolyte decomposition (Fig. 1b, left panel). No distinct position change of the graphite (002) peak is observed in the X-ray diffraction (XRD) pattern (Fig. 1b), suggesting a constant graphite $d$(002) spacing (0.336 nm, Fig. 1c) caused by the blocked PF$_6^-$ anion intercalation by the tick, non-uniform CEI as oxidative product of electrolyte (5.7 to 8.1 nm, Fig. 1c). To tackle this issue, the terminal methyl group (-CH$_3$) of the EA molecule was functionalized to an electron-withdrawing -CF$_3$ group, thereby enhancing both the Li metal compatibility and anti-oxidative stability (Fig. 1d). The as-obtained 1.2 M LiPF$_6$ salt in 2,2,2-trifluoroethyl acetate (TFEA) electrolyte delivers nonflammable properties with a SET of 0 s (inset of Fig. 1d). Furthermore, the TFEA-based electrolyte facilitates a reversible anion intercalation into/de-intercalation from the graphite host, as evidenced by the multiple voltage plateaus on the charge-discharge curve with a CE of 53.8% (the left panel of Fig. 1e) and redox peaks during the initial cyclic voltammetry (CV) curve (Supplementary Fig. 1a). Notably, the $d$(002) diffraction peak of graphite cathode progressively shifts to 23.9° (i.e., an interlayer spacing of 0.373 nm) upon charging to 5.2 V, whereas for the fully discharged state, the appearance of a distinct shoulder peak at lower angle implies incomplete PF$_6^-$ extraction or graphite expansion caused by solvent co-intercalation (the right panel of Fig. 1e and Supplementary Fig. 2). This phenomenon was verified by the irreversible lattice structure change of the distorted graphite cathode after 1st cycle, even though a thinner (-2.8 nm) and more uniform CEI is formed compared to the EA-based electrolyte (Fig. 1f). As revealed by the in-depth X-ray photoelectron spectroscopy (XPS) analysis, the graphite cathode after tested in EA electrolyte displays increasing amounts of organic species including C-O, C = O, ROCO$_2$Li and Li$_2$CO$_3$ (Supplementary Fig. 3a), as the accumulated decomposition products of EA solvent. Besides, large amounts of LiF and Li$_x$PO$_y$F$_z$ species are observed in the core F 1$s$ spectrum due to the LiPF$_6$ decomposition (Supplementary Fig. 4a). In contrast, the TFEA electrolyte achieves reduced amount of these components for the cycled cathode (Supplementary Figs. 3b and 4b), confirming alleviated side reactions between the electrolyte and graphite electrode.

Subsequently, the degree of fluorination was tuned by transitioning from -CF$_3$ groups to -CHF$_2$, resulting in a more stable solvent, DFEA (Fig. 1g). The asymmetric -CHF$_2$ group contains a local dipole, which enhances Li$^+$ solvation and reduces PF$_6^-$ coordination compared to the symmetric -CF$_3$ group, as discussed later. The commonly used fluoroethylene carbonate (FEC, 10 wt%) was added to the 1.2 M LiPF$_6$ salt in DFEA electrolyte (Supplementary Fig. 5), henceforth referred to as DFEA-based electrolyte. In addition to inheriting the high non-flammability from the TFEA (inset of Fig. 1g), the DFEA-based electrolyte enables a reversible anion de-/intercalation process on the graphite host (Supplementary Fig. 1b, c) with an improved CE of 82.0% (the left panel of Fig. 1f). The interlayer spacing of the graphite cathode expands to 0.370 nm at the fully charged state, and reversibly reverts to 0.337 nm upon discharging to 3.0 V (Fig. 1h and Supplementary Fig. 2). This illustrates that the application of DFEA solvent efficiently precludes solvent co-intercalation. The transmission electron microscope (TEM) image shows that the high resistance of DFEA to oxidation contributes to an in situ construction of a uniform and thin (-1.4 nm) CEI with reduced resistance for ion migration (Fig. 1i). The uniformity stability of the formed CEI was further verified by the XPS depth profiling, showing no obvious evolution in the LiF and Li$_x$PO$_y$F$_z$ contents as

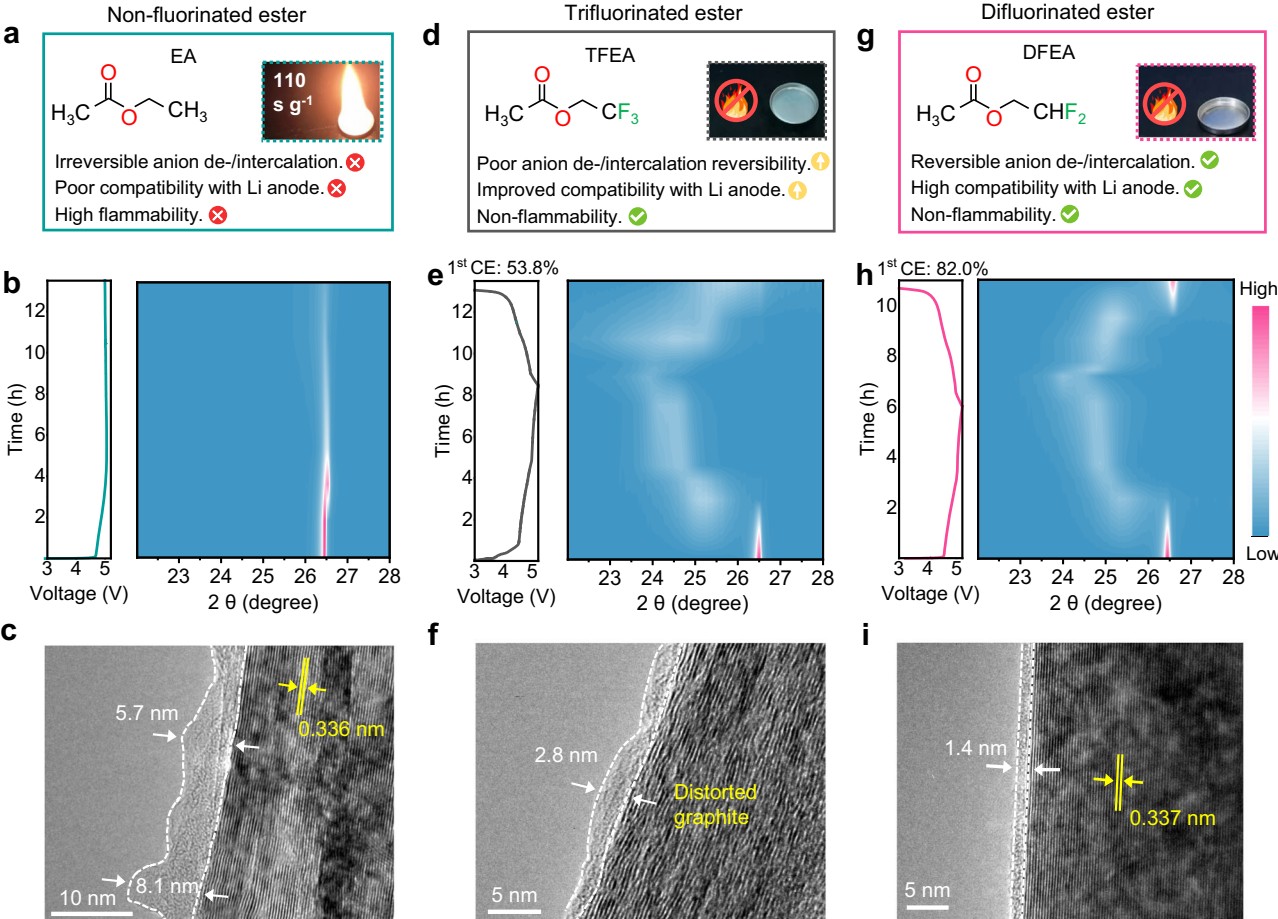

**Fig. 1 | Investigation of esters on anion-intercalation.** Molecule structure and characteristics of (**a**) EA, (**d**) TFEA and (**g**) DFEA. Combustion tests for electrolyte samples are shown in the insets. Intensity contour maps obtained from the ex-situ XRD patterns of Li||graphite cells during initial charge-discharge processes at 20 mA g$^{-1}$ using (**b**) EA, (**e**) TFEA and (**h**) DFEA (with 10 wt% FEC) electrolytes. The corresponding voltage-testing time curves are shown on the left panels. TEM images of the graphite cathodes after the 1st cycle in (**c**) EA, (**f**) TFEA, and (**i**) DFEA (with 10 wt% FEC) electrolytes.

the etching time was increased (Supplementary Fig. 4c). In addition, amounts of products derived from solvent decomposition was significantly reduced (Supplementary Fig. 3c). All these verify the superiority of DFEA in developing durable AILMBs.

## Electrochemical performance of the anion-intercalation Li metal batteries

Li||graphite AILMBs were assembled to assess their cycling performance under a high cycling current density of 1 A g$^{-1}$ (corresponding to 10 C, 1 C = 100 mAh g$^{-1}$ based on the mass of graphite active material) at 25 °C. Li||LFP LMBs using a commercial electrolyte of 1.2 M LiPF$_6$ salt in ethylene carbonate (EC)/diethyl carbonate (DEC) (denoted as EC/DEC electrolyte) were used for performance comparison (Fig. 2a). It is seen that the Li||LFP cell exhibits a 47.6% capacity retention after 3000 cycles (Supplementary Fig. 6), associating with a huge polarization increase upon cycling (Supplementary Fig. 7a). The AILMB with TFEA electrolyte delivers an initial discharge capacity of ~85.7 mAh g$^{-1}$, which gradually declines to ~24.1 mAh g$^{-1}$ at the 500$^{th}$ cycle (Fig. 2a, b), accompanied by a noticeable growing polarization (Supplementary Fig. 7b) and low CE. This short lifetime is primarily caused by structural deterioration of graphite cathode (Supplementary Fig. 8a, seen from the wrinkled thin sheets) caused by solvent co-intercalation. In sharp contrast, the DFEA-based electrolyte effectively alleviates the rise in cell polarization (Supplementary Fig. 7c), ensuring the preservation of voltage platforms with minimal deviation upon cycling. Furthermore, the average CE is as high as ~99.7% over 10,000 cycles, benefiting from

the protective effect of interphases on both the graphite cathode and the Li metal anode. Notably, the cycled graphite electrode displays a well-maintained structural integrity with laminar microstructure (Supplementary Fig. 8b). Consequently, the AILMB operates stably for 10,000 cycles with a capacity retention of 88.0% and negligible capacity fade of 0.00128% per cycle. To the best of our knowledge, this work shows improved fast-cycling stability compared to LMBs based on LFP or NCM811 cathodes and other AILMBs (Supplementary Table 1).

Furthermore, cells employing the DFEA-based electrolyte demonstrate an exceptional rate performance (Fig. 2c and Supplementary Fig. 9a), with retentions as high as ~93.0%, ~89.7%, and ~83.4% of the maximum capacity at 0.5 A g$^{-1}$ when the cycling rate increases to 6 A g$^{-1}$, 7 A g$^{-1}$, and 8 A g$^{-1}$, respectively (Fig. 2d). This remarkable ultra-fast rate capability suggests that the PF$_6^-$ anion de-/intercalation processes from/into the graphite cathode is facile and highly reversible in the DFEA-based electrolyte. In contrast, the TFEA electrolyte provides rapid capacity decay at current densities higher than 1 A g$^{-1}$, associated with an increased cell polarization (Supplementary Fig. 9b). Although LFP-based LMB is capable of achieving higher capacities at low current rates (i.e., 0.1–1 A g$^{-1}$), the reversible capacity is much inferior to that of AILMB when the current density exceeds 1.5 A g$^{-1}$, with capacity retention of only 39% and 36.5% at 7 A g$^{-1}$ and 8 A g$^{-1}$, respectively (Fig. 2c, d). The rapid capacity decay is ascribed to the growing cell polarization with increased current density (Supplementary Fig. 9c). The AILMB delivers a substantially shorter charging time of ~5.5 min to

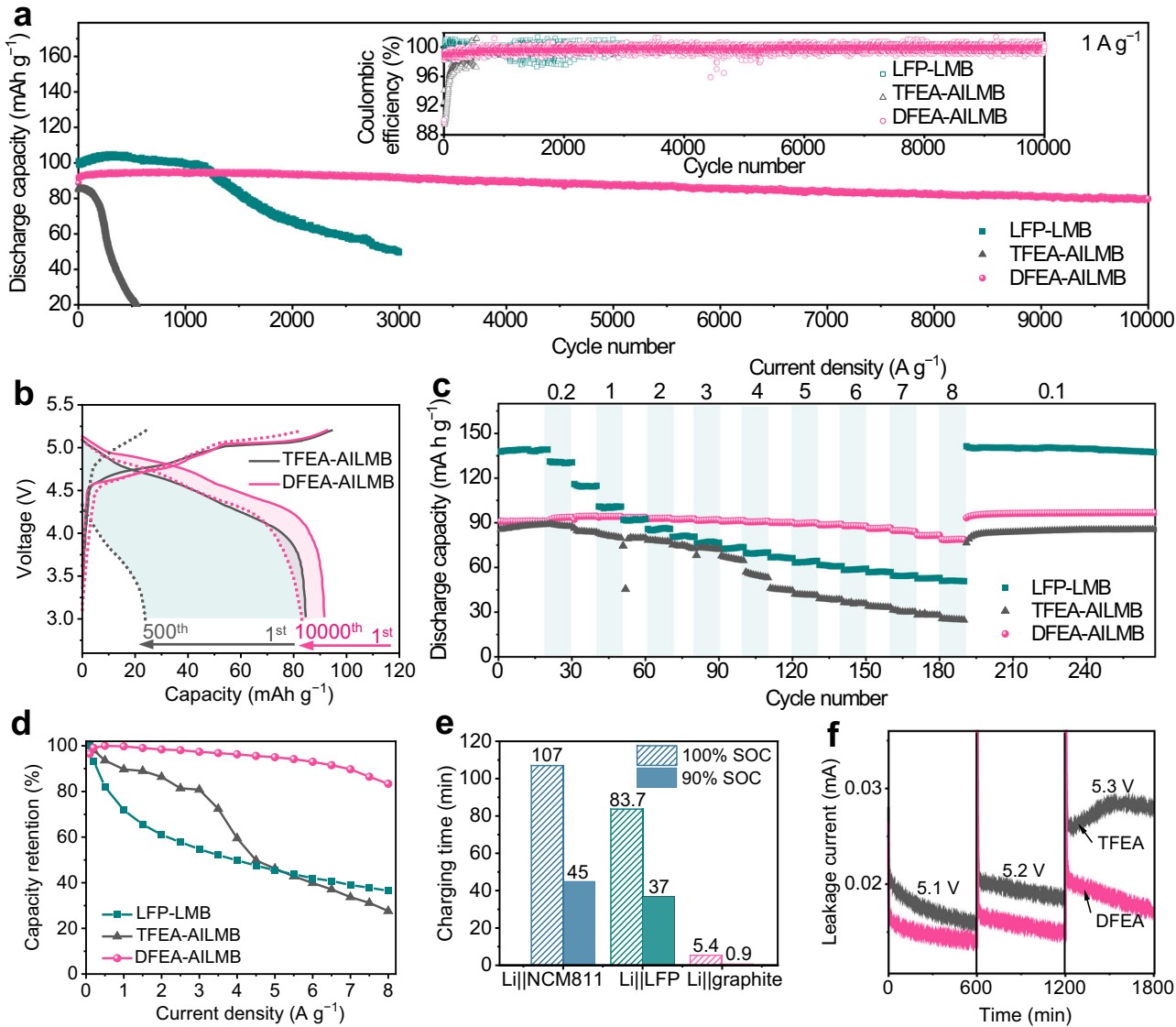

**Fig. 2 | Electrochemical performance of the anion-intercalation Li metal batteries. a** Long-term cycling performance of Li||graphite AILMBs (using TFEA and DFEA-based electrolytes) and Li||LFP LMB (using EC/DEC electrolyte) at 1 A g⁻¹ after three activation cycles at 20 mA g⁻¹. Inset is the Coulombic efficiency. **b** Typical charge-discharge curves of AILMBs using TFEA and DFEA-based electrolytes. **c** Rate performance and (**d**) corresponding capacity retention at various current densities from 100 mA g⁻¹ to 8 A g⁻¹ for AILMBs and LFP-based LMBs. **e** Comparisons of charging time for our AILMB and LMBs based on the LFP and NCM811 cathodes, at 90% SOC and 100% SOC, respectively. **f** Potentiostatic profiles of AILMBs with TFEA and DFEA-based electrolytes maintained at 5.1, 5.2, and 5.3 V for 10 h, subsequently.

reach a 100% state-of-charge (SOC, i.e., the maximum reversible capacity), compared to ~1.4 h and ~1.8 h required for the LFP-based and NCM 811-based LMBs (Supplementary Fig. 10), respectively. The charging time of this AILMB can be further reduced to within 1 min when reaching a 90% SOC (Fig. 2e). These results suggest that the anion-intercalation cathode chemistry effectively enhances the high-rate capability of LMBs, which is among the best performance reported for fast-charging LMBs (Supplementary Table 2).

The parasitic reactions on the graphite surface of AILMBs at charged state were further examined by potentiostatic tests (Fig. 2f). The cells were pre-cycled at 20 mA g⁻¹ for three cycles, followed by charged to and maintained at a constant voltage of 5.1, 5.2, and 5.3 V for 10 h, subsequently. Notably, as the voltage is raised, the cell with TFEA electrolyte exhibits much higher leakage currents compared to that with the DFEA-based electrolyte. The AILMB with TFEA-based electrolyte displays a leakage current of ~0.03 mA at 5.3 V, mainly attributed to the serious side reactions between graphite cathode with co-intercalated solvent. In contrast, the DFEA-based electrolyte effectively

stabilizes the graphite cathode even at 5.3 V. Above excellent long-term durability and ultrafast-cycling capability infer that the DFEA-based electrolyte greatly facilitates the electrode reaction kinetics in AILMBs.

## Physicochemical properties and coordination chemistry of electrolytes

Density functional theory (DFT) simulations were conducted to assess the highest occupied molecular orbital (HOMO) and the lowest unoccupied molecular orbital (LUMO) energies of EA, TFEA, and DFEA solvent molecules with various degree of fluorination. As depicted in Fig. 3a, TFEA (−8.1 eV) and DFEA (−8.0 eV) exhibit reduced HOMO energy levels compared to EA (−7.7 eV), suggesting superior anti-oxidative resistance due to the incorporation of electron-withdrawing fluorine atoms into the solvent structure[25]. This result was further confirmed by the anti-oxidative stability of electrolytes evaluated via linear sweeping voltammetry (LSV) on a Pt electrode. As expected, compared with the EA electrolyte with an oxidation potential of 4.6 V

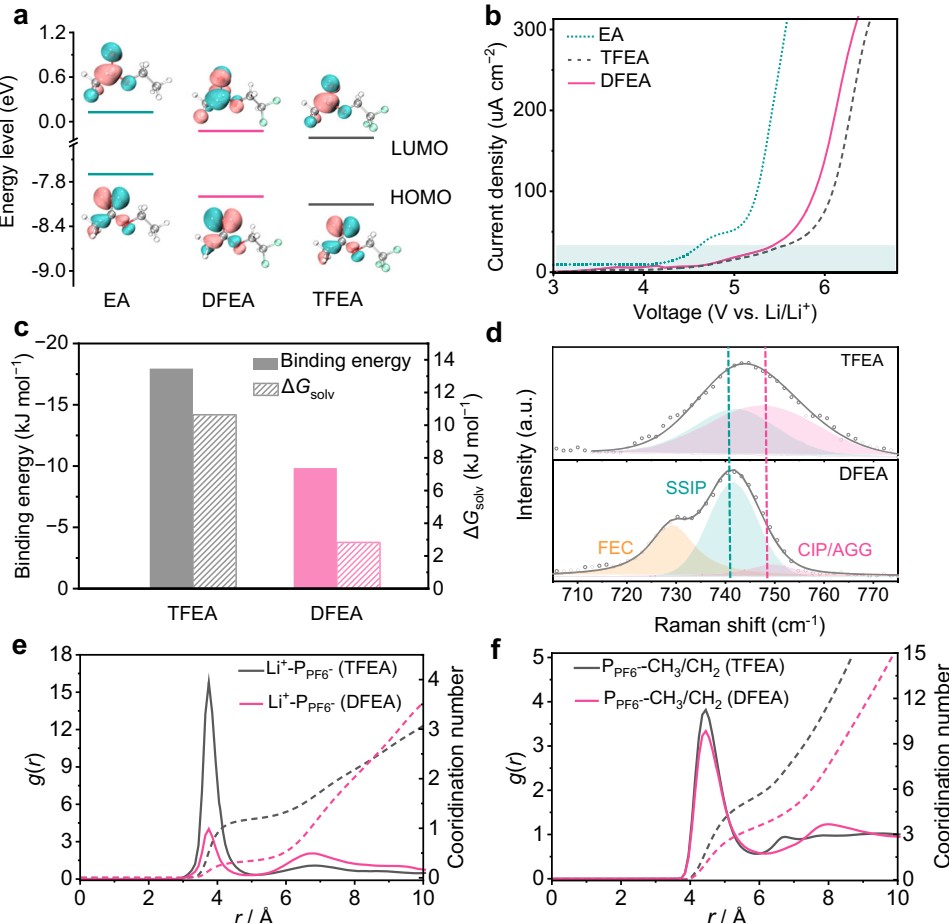

**Fig. 3 | Electrolyte properties and coordination chemistry. a** LUMO and HOMO energy values of the EA, DFEA, and TFEA solvent molecules. The molecular structures and corresponding visual LUMO and HOMO geometry structures are shown in the insets. Gray, white, red, and green balls represent carbon, hydrogen, oxygen, and fluorine atoms, respectively. **b** LSV curves of the electrolytes at a scan rate of 5 mV s$^{-1}$, employing a Pt foil as the working electrode and Li foil as the counter and reference electrodes. **c** Solvation energy ($\Delta G_{sol}$) values of electrolytes based on TFEA and DFEA solvents, and binding energies of PF$_6^-$ anion with TFEA and DFEA molecules. **d** Raman spectra of TFEA and DFEA-based electrolytes. **e** Li$^+$ and (**f**) PF$_6^-$ RDF obtained from MD simulations of TFEA and DFEA-based electrolytes. Solid lines represent g(r) while dashed lines represent coordination number.

vs. Li/Li$^+$, both TFEA and DFEA-based electrolytes remain stable up to 5.5 V, satisfying the need of high-voltage anion-intercalation cathode (Fig. 3b). Meanwhile, TFEA and DFEA also display a decline of LUMO levels (−0.22 eV and −0.13 eV, respectively) compared with EA, benefiting their preferential reduction on Li metal anodes to form robust SEI layers. To verify the ion-solvent coordination environment in electrolytes, the electrostatic potential (ESP) distribution of solvent molecules (Supplementary Fig. 11) and binding energies of both Li$^+$-solvent complexes and PF$_6^-$-solvent complexes were calculated by DFT (Fig. 3c and Supplementary Fig. 12a–c). It is seen that the DFEA molecule exhibits stronger binding energy with Li$^+$ (−101.27 kJ mol$^{-1}$ vs. −96.46 kJ mol$^{-1}$) but weaker binding energy (−9.84 kJ mol$^{-1}$ vs. −17.95 kJ mol$^{-1}$) with PF$_6^-$ compared to the TFEA. This is consistent with the ESP results (Supplementary Fig. 11), suggesting a weakened DFEA-PF$_6^-$ interaction which reduces the corresponding anion de-solvation kinetic barriers and suppresses the co-intercalation of solvents during the charging process of AILMB. In addition, the DFEA-Li$^+$ interaction is enhanced due to the existence of local dipole on the -CHF$_2$ (Supplementary Fig. 12d–h), which is similar to the recent report by Bao et al. [3]. Solvation energy ($\Delta G_{sol}$) values of the electrolytes were further determined using a homemade H-type cell to investigate the solvation structures (Supplementary Fig. 13)[26]. The DFEA-based electrolyte exhibits a much lower $\Delta G_{sol}$ (2.83 kJ mol$^{-1}$) compared to the TFEA electrolyte (10.64 kJ mol$^{-1}$, Fig. 3c), further confirming the stronger

coordination between Li ions and DFEA, which promotes the Li salt dissociation and thus gives rise to a higher ionic conductivity of DFEA-based electrolyte (7.2 mS cm$^{-1}$ at 25 °C) than the TFEA-based electrolyte (2.5 mS cm$^{-1}$ at 25 °C). Moreover, the Li$^+$ transference number of DFEA-based electrolyte (0.49, Supplementary Fig. 14 and Supplementary Table 3) is quite close to 0.5, which is beneficial to balance the active ions in the AILMBs.

Raman vibrational spectroscopy was applied to get an in-depth understanding of the electrolyte solvation structures (Fig. 3d). Two peaks at approximately 741 and 749 cm$^{-1}$ are attributed to the solvent-separated ion pair (SSIP, i.e., uncoordinated PF$_6^-$) and the contact ion pairs (CIPs, i.e., PF$_6^-$ ions interacting with one Li$^+$ ion)/aggregates (AGGs, i.e., PF$_6^-$ ions interacting with two or more Li$^+$ ions), respectively[27]. The peak at around 729 cm$^{-1}$ is assigned to the FEC (Supplementary Fig. 15). Compared with the TFEA-based electrolyte, the weaker CIP/AGG peak for DFEA-based electrolyte indicates that PF$_6^-$ anions predominantly exist in the form of SSIP, enhancing the fast ion transport. This can be further validated by molecular dynamics (MD) simulations. The MD snapshot in Supplementary Fig. 16a reveals that TFEA electrolyte displays a CIP/AGG-rich structure, where 1.22 PF$_6^-$ coordinate to one Li$^+$ based on the radial distribution functions result (Fig. 3e). In contrast, the difluoro DFEA facilitates a SSIP-rich electrolyte structure (Supplementary Fig. 16b), with 0.35 PF$_6^-$ coordinating to one Li$^+$ (Fig. 3e). Moreover, it is seen that the solvated PF$_6^-$ is

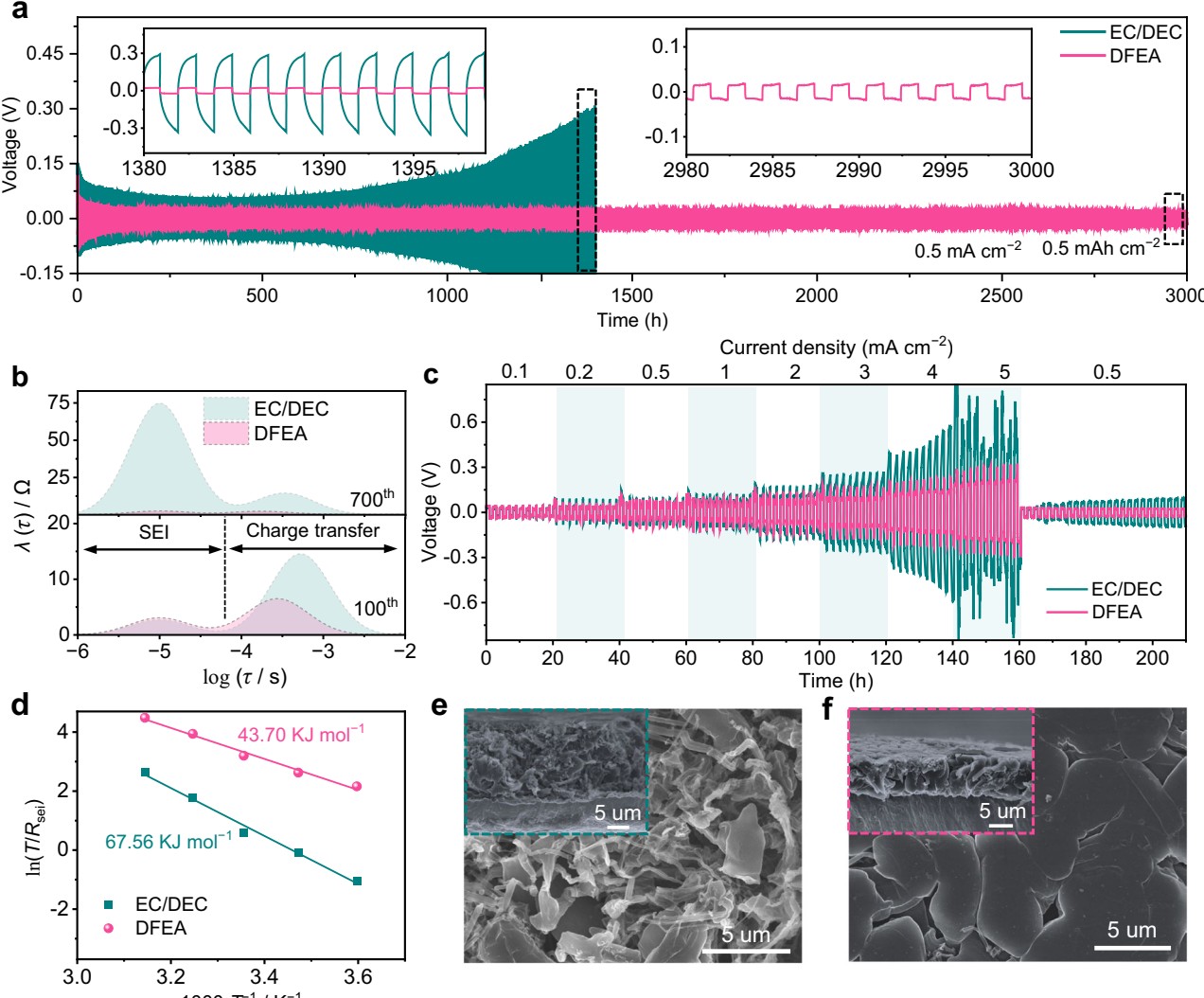

**Fig. 4 | Lithium plating/stripping behavior in different electrolytes. a** Voltage profiles of Li||Li symmetric cells employing EC/DEC and DFEA-based electrolytes at 0.5 mA cm$^{-2}$ with a cutoff capacity of 0.5 mAh cm$^{-2}$. **b** The corresponding DRT results at the 100$^{th}$ (the upper panel) and 700$^{th}$ (the lower panel) cycle. **c** Voltage profiles of Li||Li symmetric cells under different rates. **d** Activation energies of $R_{sei}$

derived from Nyquist plots using EC/DEC and DFEA-based electrolytes. Top and cross-sectional (shown as insets) FE-SEM images of the Li deposition obtained by plating 1 mAh cm$^{-2}$ Li on Cu substrate at 0.2 mA cm$^{-2}$, using the Li||Cu cells with (**e**) EC/DEC and (**f**) DFEA-based electrolytes.

distinctly coordinated with the -CH$_3$/CH$_2$ groups on the fluorinated solvent molecules, confirming the synergetic solvation environment of PF$_6^-$ (Fig. 3f). This result further verifies the balanced solvation affinity and salt dissociation of DFEA-based electrolyte, which is expected to achieve rapid de-solvation and fast ion migration of both PF$_6^-$ and Li$^+$ ions.

## Interfacial compatibility between electrolyte and the Li metal anode

The Li plating/stripping cycling behavior was investigated using Li||Li symmetric cells at a constant current of 0.5 mA cm$^{-2}$ (Fig. 4a). The cell employing EC/DEC electrolyte exhibits a substantial increase in overpotential upon cycling (302 mV at 1400 h), in stark contrast to the lower overpotential and extended lifespan (16.5 mV at 3000 h) achieved in the DFEA-based electrolyte (inset in Fig. 4a). Electrochemical impedance spectra (EIS) of symmetric Li||Li cells were conducted upon cycling to monitor the interfacial resistance, and thus, the distribution of relaxation times (DRT) analysis was derived. The peak located at ~10$^{-6}$ to 10$^{-4}$ s represents the SEI, while the peak at ~10$^{-4}$ to 10$^{-2}$ s is related to transfer process[28,29]. It is seen that the integrated area

of these two peaks for the cell using EC/DEC electrolyte increases remarkably with cycling (insets in the Fig. 4b and Supplementary Fig. 17), which is attributed to the thickening of highly resistive SEI on the Li metal and thus leading to rapid cell failure. On the contrary, the cell with DFEA-based electrolyte exhibits much smaller integral area values with slight variations as cycling, primarily due to the formation of a stable and highly conductive SEI against detrimental parasitic reactions (Fig. 4b and Supplementary Fig. 17). Moreover, the ability of the DFEA-based electrolyte to stabilize Li metal becomes more pronounced when increasing the plating/stripping current densities. At an improved current density of 1 mA cm$^{-2}$ with a cycling capacity of 1 mAh cm$^{-2}$, a reduced lifespan is observed for the symmetric cell with EC/DEC electrolyte, with the overpotential rising to 777 mV after only 750 h (Supplementary Fig. 18). Impressively, DFEA-based electrolyte exhibits much smaller overpotential and negligible fluctuation upon repeated plating/stripping processes (-80.0 mV at 1660 h). Moreover, upon further increasing the plating/stripping areal capacities, the Li||Li cells employing DFEA-based electrolyte still demonstrate significant improvements in the cycling capability with areal capacities of 3 mAh cm$^{-2}$ (930 h and 300 h at 1 mA cm$^{-2}$ and 3 mA cm$^{-2}$, respectively)

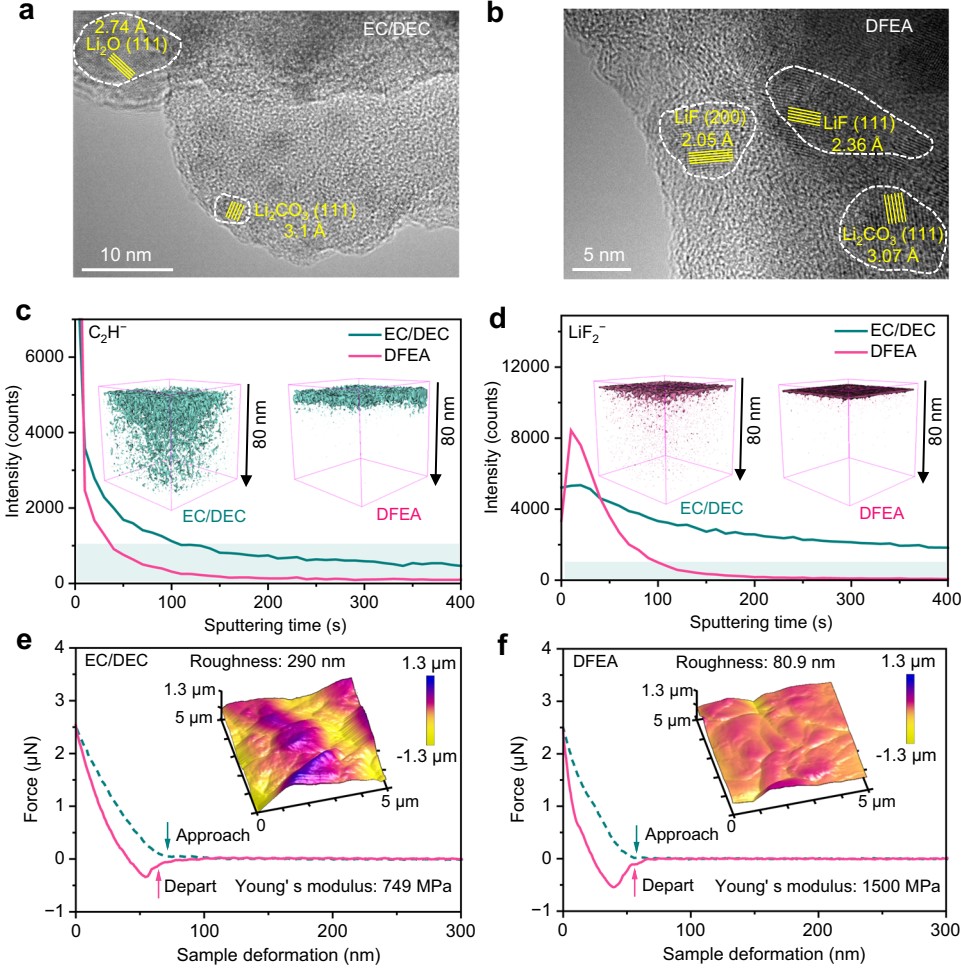

**Fig. 5 | Characterizations of the SEI on Li metal surface.** TEM image of the SEI shell formed by plating/stripping Li on a Cu grid using (**a**) EC/DEC and (**b**) DFEA-based electrolytes. Depth profiles of (**c**) $C_2H^-$ and (**d**) $LiF_2^-$ in the SEI, obtained from the TOF-SIMS. Three dimensional (3D) spatial distributions of (**c**) $C_2H^-$ and (**d**) $LiF_2^-$ are displayed in the insets. Force-displacement curves of the SEI derived from (**e**) EC/DEC and (**f**) DFEA-based electrolytes. The corresponding 3D AFM scanning images of the SEI are shown in the insets.

and $4\,mAh\,cm^{-2}$ ($730\,h$ and $500\,h$ at $1\,mA\,cm^{-2}$ and $2\,mA\,cm^{-2}$, respectively; Supplementary Fig. 19). These results represent a compelling advancement comparable with reported electrolytes (Supplementary Table 4). The Li||Li cell with DFEA-based electrolyte consistently exhibits low and stable overpotential within a wide range of current densities from 0.5 to $5\,mA\,cm^{-2}$, while substantial voltage fluctuation emerges in the cell using EC/DEC commercial electrolyte, especially when increasing the current density to $5\,mA\,cm^{-2}$ (Fig. 4c). This further confirms that the DFEA-based electrolyte enables fast Li$^+$ migration and robust SEI formation.

EIS measurements of the Li||Li cells after 20 cycles were conducted at various temperatures, and subsequently fitted by an equivalent circuit to calculate the activation energy ($E_a$) values (Supplementary Fig. 20 and Supplementary Table 5). It is seen that $E_a$ values of SEI ($R_{sei}$) and charge transfer ($R_{ct}$) resistances of the cell using DFEA-based electrolyte are much lower than that using EC/DEC electrolyte ($43.70$ vs. $67.56\,kJ\,mol^{-1}$, Fig. 4d; and $48.19$ vs. $68.06\,kJ\,mol^{-1}$, Supplementary Fig. 21, respectively). The former indicates that the SEI derived from DFEA-based electrolyte is advantageous for rapid Li$^+$ transport, meanwhile the later verifies that the DFEA-based electrolyte facilitates Li$^+$ de-solvation from the ion pairs. The stability of the electrolytes toward Li metal anodes was further estimated by the average Li plating/stripping Coulombic efficiency ($CE_{avg}$) of Li||Cu asymmetric cells[30]. The $CE_{avg}$ of DFEA-based electrolyte (99.0%) is dramatically higher

than the cell using EC/DEC electrolyte (81.4%, Supplementary Fig. 22). This phenomenon is associated with the Li deposition morphology on Cu foil presented by field emission scanning electron microscope (FE-SEM). As observed from Fig. 4e, the EC/DEC electrolyte shows a highly loose and dendritic Li deposition structure with a thickness of $18.7\,\mu m$, far exceeding the theoretical value (about $4.85\,\mu m$) and leading to the low $CE_{avg}$. In comparison, the DFEA-based electrolyte contributes to a compact Li deposit as aggregated large bulks, with a thickness (approximately $6.8\,\mu m$) quite close to the theoretical value (Fig. 4f). This dendrite-free morphology is beneficial for achieving high $CE_{avg}$ and long Li cycling stability.

The microstructures of SEI shells onto a Cu grid from Li||Cu cells (after 20 cycles and finishing with a charge process) were characterized via TEM. As displayed in Fig. 5a, for the SEI forming in the presence of EC/DEC-based electrolyte, $Li_2O$ ((111) plane, $2.74\,Å$) and $Li_2CO_3$ ((111) plane, $3.1\,Å$) nanoparticles dispersing within an amorphous SEI matrix mainly resulting from the decomposition of EC/DEC solvents. Upon cycling in the DFEA-based electrolyte (Fig. 5b), in contrast, the lattice spacing of 2.05 and $2.36\,Å$ are well matched with the (200) and (111) crystal planes of LiF nanoparticles. This LiF-rich SEI is believed to stabilize the Li|electrolyte interphase, and promote uniform Li plating during cycling. Time-of-flight secondary ion mass spectrometry (TOF-SIMS) characterizations were further performed to verify the composition/structural evolution of the SEI. It is seen that in the EC/DEC

electrolyte, $C_2H^-$ (mass charge ratio ($m/z$) = 25, as fragments of the organic products, Fig. 5c) and $LiF_2^-$ ($m/z$ = 45, fragment of LiF, Fig. 5d) signals were clearly observed throughout the 400 s-depth profiling, suggesting the formation of a thick SEI with the layer thickness exceeding 80 nm. However, the intensity of $C_2H^-$ and $LiF_2^-$ almost disappeared in DFEA-based electrolyte after sputtering for 35 s and 90 s, respectively. This indicates that the DFEA-based electrolyte contributes to a thinner SEI (around 18 nm) enriched in LiF, facilitating both rapid transport and uniform deposition of Li⁺. The roughness and thickness of the SEI were examined by atomic force microscopy (AFM). As demonstrated in the inset of Fig. 5e, the plated Li in EC/DEC electrolyte exhibits a rough surface with an average roughness of ~290 nm, considerably higher than the value of ~80.9 nm in the DFEA-based electrolyte (inset of the Fig. 5f). In addition, the Young's modulus of the SEI derived from the DFEA-based electrolyte (1500 MPa, Fig. 5f) is much higher than that formed in the EC/DEC electrolyte (749 MPa, Fig. 5e), mainly attributes to the abundant LiF species with high shear modulus (~55.1 GPa, nearly 11 times higher than that of Li metal[31]) as the decomposition product of both $LiPF_6$ salt and fluoride solvent in the SEI. This robust LiF-enriched SEI in the DFEA-based electrolyte is advantageous for improving the interfacial energy and durability against the dramatic volume change during cycling[32], thus resulting in the superior Li plating/stripping performance in Fig. 4. Notably, it is the combined effect of FEC and DFEA that endows the Li||Li cell with long-term cycling stability (Supplementary Fig. 23a, b). Both FEC and DFEA solvent contribute to the LiF-rich SEI on the Li metal, while the FEC addition effectively suppress the excessive decomposition of DFEA solvent (Supplementary Figs. 24 and 25). Furthermore, it is the DFEA solvent, rather than the FEC addition, that supports the reversible cathode reaction (Supplementary Fig. 23c, d).

## Low-temperature performance and battery safety evaluation

The ionic conductivities of the EC/DEC and DFEA-based electrolytes were compared to verify the feasibility of DFEA-based electrolyte in developing low-temperature AILMBs. As expected, the DFEA-based electrolyte exhibits high ionic conductivity (0.1–10.9 mS cm⁻¹) across a wide range of temperature (−60 to +60 °C, Supplementary Fig. 26), mainly due to its wide liquid range (as seen from the differential scanning calorimetry (DSC) measurement in Fig. 6a, where no phase change is observed within the temperature range from −80 °C to +90 °C) and balanced solvation affinity. In contrast, the EC/DEC electrolyte experiences a sudden decline in the ionic conductivity as the temperature drops below −20 °C, attributing to the electrolyte solidification which can be validated by the two distinct endothermic peaks at around −2.4 °C and −17.2 °C in the DSC curve (Fig. 6a). The low-temperature cycling performance of Li|DFEA-based electrolyte|graphite AILMB and Li|EC/DEC electrolyte|LFP LMB were assessed at −20 °C, employing the constant-current-constant-voltage (CCCV) model for charging process with a current density of 100 mA g⁻¹, followed by discharging at 500 mA g⁻¹. (Fig. 6b). The AILMB yields a remarkable cycle stability with a remaining capacity of ~80 mAh g⁻¹ over 3000 cycles without capacity degradation, the low-polarization voltage curves indicating a fast and reversible reaction kinetics of $PF_6^-$ and Li⁺ at low temperature (Supplementary Fig. 27a). Even at an ultra-high discharge current density of 8 A g⁻¹ (i.e., 80 C) under −20 °C, the AILMB retains 87.7% of its maximum capacity at 0.1 A g⁻¹ (Supplementary Fig. 28). The conventional Li||LFP LMB, however, delivers a low reversible capacity of only ~34.0 mAh g⁻¹ in the initial cycle (Supplementary Fig. 27b), and subsequently suffers from battery failure within 4 cycles due to electrolyte freezing (Inset of Fig. 6b). It is noted that even changing the conventional carbonate-based electrolyte to ether-based electrolytes (Supplementary Fig. 29), the Li||LFP LMB still exhibits much lower reversible capacity in comparison with AILMB using DFEA-based electrolyte at −20 °C. Above findings indicate that the fast migration/reaction kinetics of both cations and anions in

AILMBs is pivotal in enhancing low-temperature applicability of LMBs. Moreover, the AILMB with DFEA-based electrolyte was cycled at various temperatures, as displayed in Fig. 6c. The AILMB retains 97%, 89%, 81%, 72%, 68%, 61%, and 51.5% of its RT capacity when cycled at 10, 0, −10, −20, −30, −40 and −50 °C, respectively. The clear discharge plateaus from charge/discharge curves at −50 °C illustrate the fast electrode reaction kinetics (Supplementary Fig. 30). Notably, when charging at RT followed by discharging at −65 °C, the cell still retains ~63% of its RT capacity (inset of Fig. 6c). Furthermore, the capacity of the AILMB with DFEA-based electrolyte successfully recovers to 100% and 98.3% of its RT capacity when the testing temperature is reverted to 25 and 45 °C, respectively, reflecting the applicability of this battery system under wide temperature range. Besides, critical parameters (e.g., the capacity ratio between the anode and cathode (N/P ratio)) in cell evaluation should be reduced for practical considerations[33]. It is seen that the AILMB exhibits a high capacity retention of ~93.0% with minimal cell polarization throughout 500 cycles under N/P ratio of ~5.7 (Supplementary Fig. 31a), and it can sustains 140 cycles with ~90.7% retainable capacity under N/P ratio of ~2.0 (Supplementary Fig. 31b).

Single-layer pouch cells comprising graphite (9 mg cm⁻²) or LFP (8 mg cm⁻²) cathodes, and Li foil anodes with a thickness of 50 μm, were constructed to assess the cell safety and reliability under abusive conditions. The anion-intercalation Li||graphite pouch cell employing the DFEA-based electrolyte exhibits excellent cycling performance with 92.2% capacity retention after 300 cycles (charge at 1 C and discharge at 2 C, Fig. 6d), in stark contrast to the fast capacity degradation for the Li||LFP pouch cells (Supplementary Fig. 32). Beyond this, the AIMIB pouch cell consistently powered a light-emitting diode under bending, folding and even rolling (insets in Fig. 6d), owing to the robust interphases on both cathode and anode against violent shape deformation. More importantly, the total exothermic heat generated from the delithiated LFP with EC/DEC electrolyte is as high as 461.12 J g⁻¹ (Fig. 6e), which is believed to be the origin of a succession of exothermic reactions that lead to uncontrollable battery thermal runaway[34,35]. Unexpectedly, the charged graphite cathode with DFEA-based electrolyte exhibits an endothermic peak during the DSC test (Fig. 6e). This finding is consistent with previous report[36], clearly indicating the high thermal stability of the charged $C_n(PF_6)$ cathode has been distinctively improved by the protective CEI layer derived from DFEA-based electrolyte. To further assess the safety of the DFEA electrolyte-based AILMB, an overcharge abuse test was carried out by charging the pouch cells from open circuit voltage (OCV) to 9 V at a rate of 20 mV s⁻¹ (Fig. 6f). Clearly, the current density of the LFP-based LMB increases abruptly from 6.5 V, accompanied by an elevated skin temperature of 49.9 °C at a voltage of around 7.8 V (upper-panel insets of Fig. 6f), typically triggered by the decomposition of the electrode interphases under thermal abuse. The AILMB utilizing the DFEA-based electrolyte, in contrast, exhibits superb overcharging resistance up to 9 V without obvious change in current density, as evidenced by the low skin temperature less than 35 °C (lower-panel insets of Fig. 6f). Such a high overcharge resistance is primarily attributed to the superior anti-oxidative stability of the DFEA-based electrolyte, the robustness of the protective CEI formed on the graphite cathode, as well as the intrinsic stability of the $C_n(PF_6)$ structure. Above results well-support the dependability of our AILMB under extreme working conditions.

Furthermore, a 440 mAh Li||graphite multi-layer pouch cell was packaged, with the specific energy being calculated as ~141.7 Wh kg⁻¹ (Supplementary Fig. 33 and Supplementary Table 6). Moving forward, it is imperative to conduct further research on modifying graphite cathodes (e.g., surface treatment, doping), optimizing electrolytes (e.g., anions with smaller sizes, multivalent anions with more charge numbers, and solvent with lower density), and refining engineering issues, to improve the specific energy of AILMBs without compromising their superior fast-cycling, low-temperature, and safety characteristics.

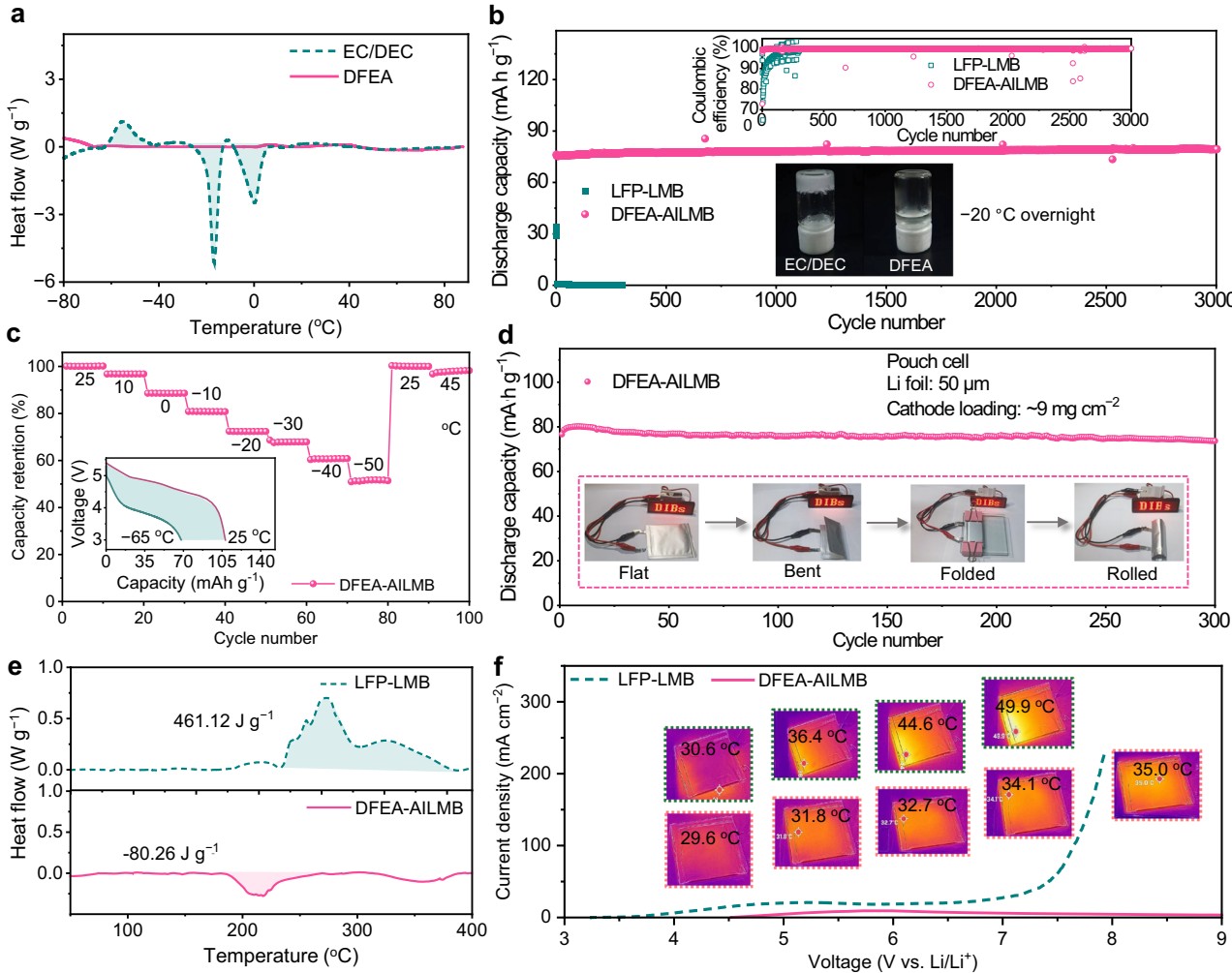

**Fig. 6 | Electrochemical performance of anion-intercalation Li metal batteries under low-temperature and abusive conditions. a** DSC cooling and heating curves of EC/DEC and DFEA-based electrolytes. **b** Long-term cycling performance of the AILMB (using DFEA-based electrolyte) and the LFP-LMB (using EC/DEC electrolyte) at −20 °C, employing the CCCV model for charging process with a current density of 100 mA g⁻¹, followed by discharging at 500 mA g⁻¹. Insets show the Coulombic efficiency (upper panel) and the optical images of the electrolytes after storage at −20 °C overnight (lower panel). **c** Capacity retentions of AILMB using DFEA-based electrolyte under various temperatures (45, 25, 10, 0, −10, −20, −30, −40 and −50 °C). Inset shows the discharge profiles when the cell is charged at RT followed by discharged at RT or −65 °C. **d** Cycling performance of the 50 μm-thick Li foil‖graphite pouch cell at a charge rate of 1 C and a discharge rate of 2 C. Optical images of the LED powered by the pouch cell under various deformations are shown in insets. **e** Heat generation of fully charged graphite and LFP cathodes together with their respective electrolytes measured by DSC. **f** LSV curves and corresponding infrared thermal imaging photographs (shown in insets) of the Li‖graphite and Li‖LFP pouch cells at a scan rate of 20 mV s⁻¹ from OCV to 9 V.

## Discussion

We developed a difluoroester-based electrolyte to realize ultra-fast, long-term cycling stability of transition metal-free AILMBs under extreme working conditions. Compared with the trifluoro counterpart, the difluoroester as solvent effectively modulates the solvation structure by attenuating the anion-solvent interactions, thereby striking a balance between the solvation affinity and salt dissociation of DFEA-based electrolyte. This not only endows rapid de-solvation and fast migration for both $Li^+$ cations and $PF_6^-$ anions to accelerate electrode reaction kinetics, but also efficiently suppresses the co-intercalation of solvent molecules to promote the cathode reaction reversibility. Our DFEA-based electrolyte simultaneously possesses high ionic conductivity, remarkable thermodynamically oxidative stability, excellent Li metal cyclability, and high safety without the risk of combustion. Morphological analysis and interphase investigations confirm the formation of highly stable protecting interphases on both graphite cathode and Li metal anode, contributing to a durability of 10,000 cycles (with negligible capacity fade of 0.00128% per cycle) at 1 A g⁻¹,

an excellent rate capability at 8 A g⁻¹ (with a retention of ~83.4% of the maximum capacity at 0.5 A g⁻¹), superior ultra-low temperature capability with ~63% of their RT discharge capacity at −65 °C, and abuse-tolerant capabilities (e.g., robustness against deformation and overcharge) for AILMBs. The electrolyte engineering proposed in this work is anticipated to expedite the re-design of Li metal batteries, enabling the adoption of cost-effective and environmental benign cathode materials, therefore advancing the development of high-rate cycling Li metal batteries under severe actual working conditions.

## Methods

### Electrolyte preparation

$LiPF_6$ (purity ≥ 99.95%) salt, FEC (purity ≥ 99.99%) solvent, EC (purity ≥ 99.99%) solvent, and DEC (purity ≥ 99.99%) solvent were purchased from Dongguan Shanshan Battery Materials Co., Ltd. EA (purity > 99.5%) solvent was obtained from Aladdin Bio-Chem Technology Co. Ltd. DFEA (purity 99%) solvent and TFEA solvent (purity 99%) were purchased from Shangfluoro. Before use, all solvents were

further dehydrated using 4 Å molecular sieves (provided by Sigma-Aldrich) for 48 h. The electrolytes were prepared by dissolving 1.2 M $LiPF_6$ salt in EA, TFEA, DFEA, and commercial EC/DEC (1:1 by volume) solvents, respectively. 10 wt% FEC was added into the 1.2 M $LiPF_6$ in DFEA electrolyte. All electrolyte preparations were conducted in an argon-filled glove box with $O_2$ and $H_2O$ levels maintained below 0.1 ppm.

## Electrolyte characterizations

The Raman spectroscopy was conducted with a Micro-laser confocal Raman spectrometer (Horiba LabRAM HR800, France) with a 532 nm laser to investigation the solvation structure. In the combustion test, 1 g electrolyte samples were poured into a stainless-steel dish, then optical photographs and movies were recorded. The melting points of the electrolytes were evaluated using differential scanning calorimetry (DSC, MDTC-EQ-M06-01). The electrolytes were sealed in stainless-steel crucible, weighed (15 μL) and subjected to DSC measurement with a ramp rate of 5 °C min$^{-1}$. The ionic conductivities of different electrolytes at various temperatures were determined from the EIS results, obtained using two symmetrically placed stainless-steel electrodes in the electrolyte. The test cells were equilibrated at each temperature for at least 1 h before EIS measurements. The electrochemical stabilities of the electrolytes were evaluated by LSV tests conducted on a three-electrode system, with a sweep rate of 5 mV s$^{-1}$. Li foil served as counter and reference electrodes, while platinum foil was used as the working electrode[37]. The onset of the oxidation current density was defined as 30 μA cm$^{-2}$ and the potential values were recorded. The solvation energy of electrolytes was evaluated using an H-type cell, where symmetrical Li metal electrodes immersed in the reference electrolyte (1.2 M $LiPF_6$-DEC) and the tested electrolyte were connected by a salt bridge (3 M $LiPF_6$-EMC). The measured open circuit potential ($E_{cell}$) determined the solvation free energy ($\Delta G_{sol}$), as illustrated below[26]:

$$E_{cell} = \frac{-\Delta G_{sol}}{F} \tag{1}$$

Where F is the Faraday constant.

To investigate the impact of the electrolyte on the long-term durability of the Li metal anode, galvanostatic cycling of Li||Li symmetric cells was conducted at current densities of 0.5 mA cm$^{-2}$ and 1 mA cm$^{-2}$, respectively, with an areal capacity of 1 mAh cm$^{-2}$. The rate capability of Li metal anode was assessed through repeated 1 h charge–1 h discharge cycles at various current densities ranging from 0.5 mA cm$^{-2}$ to 5 mA cm$^{-2}$. For activation energy ($E_a$) measurements, symmetric Li||Li cells with various electrolytes were cycled 20 times at a current density of 0.5 mA cm$^{-1}$ with an areal capacity of 0.5 mAh cm$^{-1}$. Subsequently, these cycled cells were subjected to temperature-dependent EIS measurements at 278, 283, 288, 293, and 298 K. By fitting the EIS data via an equivalent circuit, the SEI resistance ($R_{sei}$) and charge transfer resistance ($R_{ct}$) values were derived. The $E_a$ was then calculated according to the Arrhenius equation as follows[38,39]:

$$k = \frac{T}{R_{res}} A \exp\left(-\frac{E_a}{RT}\right) \tag{2}$$

where k denotes the rate constant, $T$ is the absolute temperature, $R_{res}$ corresponds to $R_{ct}$ or $R_{sei}$, A signifies the preexponential constant, and R is the standard gas constant. The CEs of Li deposition/stripping were evaluated using Li||Cu cells, based on the Aurbach's CE test protocol[30]. Initially, the Cu electrode was pre-deposited with Li metal at a capacity of 5 mAh cm$^{-2}$ at a current density of 0.5 mA cm$^{-2}$ and subsequently stripped Li to 1 V. Next, the Cu was deposited with 5 mAh cm$^{-2}$ of Li to form a Li reservoir ($Q_T$). Afterwards, the cell underwent repeated charge/discharge cycles with a capacity of 1 mAh cm$^{-2}$ ($Q_C$) for n cycles,

followed by stripping all remaining Li reservoir to 1 V ($Q_S$). The average CE ($CE_{avg}$) over n cycles can be calculated as follows[30]:

$$CE_{avg} = \frac{nQ_C + Q_S}{nQ_C + Q_T} \tag{3}$$

## Battery assembly and characterizations

Metallic Li foils with thickness of 450 μm and 50 μm were purchased from China Energy Lithium Co. Ltd (Tianjin, China). Thinner Li foils with a thickness of 20 μm was purchased from Guangdong Canrd New Energy. Graphite (SAG-R) and LFP were purchased from Shenzhen Kejing Star Technology Co. Ltd. and Shenzhen Dynanonic Co., Ltd., respectively. Acetylene black and Super P were provided by Alfa Aesar. Polyacrylic acid (PAA, MW 450,000) binder and poly(vinylidene fluoride) (PVDF, MW 1,200,000) binder were purchased from Sigma Aldrich and Arkema, respectively. N-methylpyrrolidone (NMP, purity 99.9%) was provided by Aladdin Bio-Chem Technology Co. Ltd. NCM811 particles, 2032-type coin cells, and other battery materials were provided by Guangdong Canrd New Energy. The graphite cathode slurry was prepared by mixing graphite powder as the active material, acetylene black and carbon nanotubes (CNT, TUBALL BATT NMP) as conductive agents, and PAA as binder in a mass ratio of 85: 4.5:0.5:10, using NMP as the solvent. The LFP and NCM 811 slurry were obtained by blending the active material, Super P and PVDF at a weight ratio of 80: 10: 10. These homogeneous slurries were coated onto Al foils with a doctor blade and dried at 80 °C. The dried electrodes were punched into circular sheets with a diameter of 12 mm, and the average mass loading of active material on each electrode was 2 ± 0.5 mg cm$^{-2}$. The thickness was 33 ± 4 μm for graphite electrodes and was 30 ± 3 μm for LFP and NCM 811 electrodes. The graphite electrode was further dried at 150 °C, while the LFP and NCM811 electrodes were dried at 100 °C under vacuum overnight before cell assembly. The CR2032-type coin cells, comprising the obtained electrodes and Li metals (450 μm), were fabricated in an Ar-filled glove box with $O_2$ and $H_2O$ content below 0.1 ppm. A PVDF membrane (Merck Millipore Ltd., pore size: 0.2 μm) with a diameter of 19 mm was employed as the separator, and the electrolyte/graphite ratio in each cell was set at around 60 μL mg$^{-1}$. Galvanostatic charge–discharge tests were carried out on a Neware battery testing system. The Li||graphite, Li||LFP, Li|| NCM811 cells underwent three formation cycles at a current density of 20 mA g$^{-1}$. Subsequently, the cells were cycled at a constant current of 1000 mA g$^{-1}$ within the voltage ranges of 3–5.2 V (for Li||graphite), 2.5–4.3 V (for Li||LFP), and 2.8–4.4 V (for Li||NCM811). The rate capability was assessed by varying the charge/discharge rate from 100 mA g$^{-1}$ to 8 A g$^{-1}$. Potentiostatic tests were performed on Li||graphite cells, which were charged to and maintained at constant voltages of 5.1, 5.2, and 5.3 V for 10 h each. The CV, LSV, and impedance measurements were executed using the Bio-Logic potentiostat (VMP3). CV tests of Li||graphite cells were conducted at a scan rates of 0.1 mV s$^{-1}$, while LSV tests were performed on a Ti working electrode at a sweep rate of 5 mV s$^{-1}$. Impedance measurements were carried out by applying a 10 mV potential amplitude within the frequency range from 100 kHz to 10 mHz. The DRT analysis was carried out using the DRT tools developed by Francesco Ciucci et al.[29] For low-temperature electrochemical tests, the Li||graphite or Li||LFP cells were initially activated for 100 cycles at room temperature with a current density of 1000 mA g$^{-1}$. The cells were then transferred to a temperature chamber allowed to equilibrate for 2 h to reach −20 °C, followed by cycling under a CCCV model. Specifically, the cells were charged to 5.4 V (Li|| graphite) and 4.5 V (Li||LFP), respectively, at 100 mA g$^{-1}$ and the voltage was maintained until the charging current decreased to 10 mA g$^{-1}$, followed by discharging at a constant current density of 500 mA g$^{-1}$. To evaluate the capacity retention at various temperatures, after the activation process depicted above, the cells were charged at

200 mA g$^{-1}$ (45 °C), 100 mA g$^{-1}$ (45, 25, 10, 0, −10, −20 °C), 20 mA g$^{-1}$ (−30, −40 °C) and 10 mA$^{-1}$ (−50 °C), respectively, using the CCCV mode, followed by discharging at a constant current density of 100 mA g$^{-1}$. The test at −65 °C was conducted with RT-charge at 100 mA g$^{-1}$ using the CCCV mode, followed by discharging under −65 °C at 10 mA g$^{-1}$.

Single-layer pouch cells were assembled utilizing the graphite electrode or the LFP electrode as the cathode (cathode dimension: 50 mm × 40 mm), paired with a 50 μm-thick Li foil as the anode (anode dimension: 52 mm × 42 mm). Al and Nickel strips were affixed to the sides of cathode and anode, respectively, to serve as electrode tabs. The mass loading of the cathode was 8–9 mg cm$^{-2}$. The N/P ratios were ~15.5 and ~10.6 for the Li||graphite and the Li||LFP pouch cells, respectively. A PVDF separator impregnated with 20 μL cm$^{-2}$ electrolyte was laminated and assembled into Al plastic film packages, which were then sealed under vacuum. Subsequently, the cells were aged at 25 °C for 12 h and degassed following the initial cycle. The assembled pouch cells were pre−cycled between 3–5.2 V at 20 mA g$^{-1}$. Subsequently, the pouch cells were charged at 100 mA g to 5.2 V and discharged at 200 mA g$^{-1}$. A 440 mAh Li||graphite multi-layer pouch cell with a N/P ratio of ~2.4 was packaged. The pouch cell was charged at 20 mA g$^{-1}$ using the CCCV mode, followed by discharging at a constant current rate of 20 mA g$^{-1}$. The specific energy of the AILMB was calculated as follows[40,41]:

$$E_{full} = C_{full} \times V_{cell}/(m_{total}) \qquad (4)$$

where $V_{cell}$ and $C_{full}$ are the average working voltage and the reversible capacity of the full cell, respectively, $m_{total}$ is the total weight based on the sum of current collector, cathode, anode, separator and electrolyte. The weight of the packing cell bag is excluded from the specific energy calculation due to our limited size of cell[42,43]. To assess the overcharge-safety, the pouch cells were charged from the open circuit voltage to 9 V at a scan rate of 20 mV s$^{-1}$, and the infrared thermography images of the pouch cells were captured using a FLIR ONE PRO.

**Postmortem analysis of cycled batteries**

The cycled cells were disassembled in a glove box. The electrodes were rinsed with dimethyl carbonate (DMC) solvent several times to remove residual electrolyte, and then dried in the antechamber under vacuum of the glove box. Ex-situ XRD (Rigaku D/max-2500) with Cu Kα radiation (λ = 1.5418 Å), was conducted to investigate the crystal structure changes during the initial charging and discharging processes. The evolutions in the interphases and morphologies for the cathode materials and Li metal anodes were studied using a field-emission scanning electron microscope (FE-SEM, SU8010) and a transmission electron microscope (TEM, FEI Tecnai G2 F30). X-ray photoelectron spectra (XPS) was obtained using a PHI 5000 Versa probe II spectrometer using monochromatic Al K(alpha) X-ray source. The depth values in the XPS depth profiles were estimated from the calibrated sputtering of SiO$_2$. The collected spectra were calibrated based on the C 1 s binding energy of 284.8 eV and analyzed using Multipak software. Time-of-flight secondary ion mass spectrometry (TOF-SIMS) was performed on a Nano TOF-2 instrument (ULVAC-PHI, Japan) with a 30 kV Bi$_3$ + + beam cluster primary-ion gun and Ar$^+$ beam (3 keV 100 nA) for depth profiling with a sputtering rate of 0.2 nm s$^{-1}$. The surface roughness and 3D morphologies of the cycled Li metal anodes were characterized using atomic force microscopy (AFM, Bruker Dimension Icon). To investigate the thermal stability of the charged electrodes, the cells were pre−cycled twice and finally charged to 5.2 V (Li||graphite) and 4.3 V (Li||LFP), respectively. DSC samples were prepared by scraping the dried electrode materials off the Al current collector, and 1 mg of the material was sealed in a Mettler high-pressure stainless-steel pairing with 3 μL of electrolyte. The DSC measurement was conducted from room temperature to 400 °C with a ramp rate of 5 °C min$^{-1}$.

**Computations**

The ORCA software package was used to carry out DFT calculations, with molecular geometries for the ground states optimized at the B3LYP-D3/def2-TZVP level[44]. Molecular orbitals and ESPs were analyzed using Multiwfn and visualized through VMD[45]. MD simulations were conducted here to investigate the solvation structures of above electrolytes. Interactions between ions and molecules were described based on the Optimized Potentials for Liquid Simulation All-Atom (OPLS-AA) force field. All MD simulations were performed using the LAMMPS software[46]. The simulation boxes were constructed with PACKMOL software[47]. The molar ratios were taken from experiments in this study. Each system commenced with energy minimization employing the conjugate gradient algorithm, followed by a 1 ns Brownian dynamics at 500 K to relax the system temperature and randomize the initial shape of each component. Subsequently, a 10 ns NpT equilibration at 298 K was conducted to ensure the equilibrium of salt dissociations. A 20 ns NVT production run at 298 K and then performed, with the final 5 ns reserved for data analysis. The MD snapshots were visualized using Ovito software. A velocity-Verlet integrator was employed to update the positions and forces of atoms with a timestep of 1 fs. To account for long-range interactions, a particle-particle particle-mesh method was utilized, with a global cutoff of 12 Å established for both Lennard-Jones and Coulombic interactions. Periodic boundary conditions were applied in all dimensions[48].

**Reporting summary**

Further information on research design is available in the Nature Portfolio Reporting Summary linked to this article.

## Data availability

The data that support the findings of this study are available from the corresponding author upon request.

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

## Acknowledgements

F.K. would like to acknowledge the support from National Key Research and Development Program of China (2022YFB2404500) and Shenzhen Outstanding Talents Training Fund. D.Z. would like to acknowledge the support from the Fundamental Research Project of Shenzhen (NO. JCYJ20230807111702005). Y.W. would like to acknowledge the support from National Natural Science Foundation of China (NO. 22309102) and China Postdoctoral Science Foundation (Grant No. 2022M711788).

## Author contributions

D.Z. and F.K. conceived and designed this work. Y.W. performed the experiments and wrote the manuscript. Y.G. and Y.T. carried out the computations. S.D., P.L., Y.M., X.H., X.Z., and B. L. discussed the results and participated in the preparation of the paper.

## Competing interests

The authors declare no competing interests.
