## [Peer Review File · Nature Communications]

Difluoroester solvent toward fast-rate anion-intercalation lithium metal batteries under extreme conditionsREVIEWER COMMENTS

Reviewer #1 (Remarks to the Author):

The development of high energy and safe secondary batteries are significant to expand electric vehicle industry, and the implementation of the lithium metal as the anode material is highly crucial to increase the cell-level energy density. Along with the improved reversibility of the lithium metal electrode, the high voltage graphite-based cathode was enabled in this work by DFEA-based electrolyte, which diminishes the interfacial degradation induced from typical Li-containing cathode materials. The various electrochemical performances are substantially enhanced with DFEA electrolyte, correspondingly. The characterization and the design of the solvent was properly demonstrated in this work; however, the evaluation condition and several points should be clarified to verify the effectiveness of the DFEA electrolyte. Below are comments:

- 1) The authors compare the energy density and the low temperature/high C-rate cycle performance with LFP-based LMBs, however the comparison is not fair. The state-of-art specific energy of the Li containing cathode-based LMBs exceeds the 350~400Wh/kg. What is the most significant advancement compared with those LMBs? Further, the low temperature capacity retention was compared with EC/DEC-based electrolyte for LFP and DFEA-based for graphite LMB in this work. Because the ionic conductivity of EC-based electrolyte is readily decreasing at -20 oC, this comparison is not adequate for comparative study by typical electrolyte differences. The experimental condition is too favorable to authors electrolyte. Reviewer suggests the comparative study with well-known ether-based electrolyte for LMBs, such as LiFSI-DME or LiFSI-DME/TTE based systems for Li/LFP cell.
- 2) Over 10000 cycles with the lithium metal anode are highly promising result. However, the evaluation condition highly alters the demonstrated cycle performances from the LMBs. For instance, the loading level of cathode, NP ratio, and the injected electrolyte amount are significantly influencing the cycle performances. Please verify these conditions. Because the thickness of Li electrode in this study was 450 μm , the Li reservoir is highly included in the cell. Since the thick lithium metal was comprised in this work, the capacity retention is highly extended even with the Li-consumption. Thus, the cell evaluation should be conducted at NP ratio below 2 for high specific energy system.
- 3) The electrolyte-to-capacity (E/C) ratio of the cell evaluation condition is too high. The 60 $\mu\text{L}/\text{mg}$ is flooded condition, indicating that the specific energy of LMB was below 300Wh/kg. Compared to specific energy of state-of-art lithium-ion batteries (>300Wh/kg), this evaluation condition is not meaningful for the advanced batteries system. (Joule, 2019, 3, 1094) The capacity retention from DFEA electrolyte should be reported based on the lean electrolyte condition ($E/C < 2.5$)
- 4) The electrolyte for high energy batteries should be evaluated at high areal capacity, and hence the symmetric cell evaluation should be performed over 3.0 mAh/cm²
- 5) With DFEA electrolyte, what is the specific energy demonstrated? The obtained specific energy should be stated. This concerns is highly correlated with flooded evaluation condition.

Reviewer #2 (Remarks to the Author):

This paper reports a new electrolyte solvent for anion-intercalation lithium batteries. The

authors demonstrate that 2,2-difluoroethyl acetate (DFEA) works as a good electrolyte solvent that enables highly stable cycling of the battery. It is also shown that the difluoro part is better than usual trifluoro moiety. The solvent is novel and the battery performance is impressive. However, there are some problems that must be addressed before accepted for publication.

Comments

1. The title should include some description of “anion-intercalation” because the DFEA solvent mainly contributes to the anion-intercalation reaction. There are many solvents that show better Coulombic efficiency for Li metal anode side.
2. For nomenclature, the authors refer this class of solvent (EA, TFEA, and DFEA) as carboxylates, but they are usually referred to as esters. Carboxylates are an anion with a structure of RCOO^- .
3. The solvation structures of Li^+ and EA, TFEA, and DFEA should be shown. Does the partial dipole in the difluoro moiety in DFEA contributes to the Li^+ solvation?
4. The comparison in Fig. 1 is not fair because FEC is added to only DFEA. This fact is not described in the Figure or the caption. The authors claim that the amount of FEC is small but the addition of 10 wt% is very large. Usually, additives are used less than 5 wt%. This figure misleads the readers, so it should be revised.
5. This FEC addition causes the same problems in Figs. 4-6. It is well known that FEC works as a very good additive to enable highly reversible Li plating/stripping. So, it is not clear whether the high cycling performance is derived from DFEA or FEC. Furthermore, since FEC forms LiF-rich thin SEI on Li metal, the observed SEI components in Fig. 5 may primarily be derived from FEC, not from DFEA, which significantly compromises the impact and novelty of this work.

Response to Reviewers' Comments

Reviewer #1 (Remarks to the Author):

The development of high energy and safe secondary batteries are significant to expand electric vehicle industry, and the implementation of the lithium metal as the anode material is highly crucial to increase the cell-level energy density. Along with the improved reversibility of the lithium metal electrode, the high voltage graphite-based cathode was enabled in this work by DFEA-based electrolyte, which diminishes the interfacial degradation induced from typical Li-containing cathode materials. The various electrochemical performances are substantially enhanced with DFEA electrolyte, correspondingly. The characterization and the design of the solvent was properly demonstrated in this work; however, the evaluation condition and several points should be clarified to verify the effectiveness of the DFEA electrolyte. Below are comments:

1. The authors compare the energy density and the low temperature/high C-rate cycle performance with LFP-based LMBs, however the comparison is not fair. The state-of-art specific energy of the Li containing cathode-based LMBs exceeds the 350~400Wh/kg. What is the most significant advancement compared with those LMBs? Further, the low temperature capacity retention was compared with EC/DEC-based electrolyte for LFP and DFEA-based for graphite LMB in this work. Because the ionic conductivity of EC based electrolyte is readily decreasing at -20 °C, this comparison is not adequate for comparative study by typical electrolyte differences. The experimental condition is too favorable to authors electrolyte. Reviewer suggests the comparative study with well-known ether-based electrolyte for LMBs, such as LiFSI-DME or LiFSI-DME/TTE based systems for Li/LFP cell.

Reply: Thanks for the valuable and insightful comments which greatly improve the quality of our work. We agree with the Reviewer that by employing high-energy cathode materials, such as lithium nickel cobalt manganese oxides ($\text{LiNi}_{1-x-y}\text{Co}_x\text{Mn}_y\text{O}_2$, NCM), LMBs can yield higher energy density of 350–400 Wh kg^{-1} compared with our anion-intercalation LMBs (AILMBs). Accordingly, we have deleted the comparison of energy density in Fig. 2f. Actually, our anion-intercalation LMBs offer unique merits in the following aspects:

1) First, the AILMB exhibits fast-rate operational characteristics. The AILMB delivers a substantially shorter charging time of ~5.5 min to reach a 100% state-of-charge (SOC, i.e., the maximum reversible capacity), compared to ~1.4 h and ~1.8 h required for the LFP-based and NCM 811-based LMBs (Supplementary Fig. 10), respectively. The charging time of this AILMB can be further reduced to within 1 min when reaching a 90% SOC (Fig. 2e). These results suggest that the anion-intercalation cathode chemistry effectively enhances the high-rate capability of LMBs, which is among the best performance reported for fast-charging LMBs (Supplementary Table 2). This discussion has been added on Page 9 of the Revised Manuscript.

Supplementary Table 2. Comparisons of the fast-charging capability of reported LMBs. For comparison, the unit of current densities have been standardized to mA cm⁻².

Battery configuration	Capacity retention	Current density	Charging time	Ref.
Li 1 M LiPF ₆ -EC/DEC+0.1 M LiTFA+0.1 M LiNO ₃ NCM811	~67%	6 mA cm ⁻²	6 min	Xia, Y. et al. Nat. Energy 8 , 934-945 (2023).
Li 2 M LiPF ₆ -DMC C2DP-G	~59.5%	4 mA cm ⁻²	3 min	Sabaghi, D. et al. Nat. Commun. 14 , 760 (2023).
Li 1 M LiPF ₆ -EC/DEC+ 0.25 wt.% PFPA+1 wt.% FEC Mn-rich cathode	~38%	6 mA cm ⁻²	3 min	An, K. et al. Adv. Funct. Mater. 33 , 2301755 (2023).
Li TMO LFP	~50%	~11.9 mA cm ⁻²	3 min	Wang, H. et al. Adv. Mater. 2313135 (2024).
Li 1 M LiPF ₆ -EC/DEC+1 wt.% BFA NCM622	~67%	5 mA cm ⁻²	~10 min	Li, F. et al. Angew. Chem. Int. Ed. 60 , 6600 (2021).
Li 1.2 M LiPF ₆ -EC/DEC LFP	~52%	7 mA cm ⁻²	~3 min	Performance comparison
Li 1.2 M LiPF ₆ -DFEA+10 wt.% FEC Graphite	~84%	16 mA cm ⁻²	45 s	This work

2) Besides, the DFEA-based electrolyte enables the AILMB with superior ultra-low temperature capability, retaining ~63% of its room-temperature discharge capacity at -65 °C and achieving a remarkable cycle stability at -20 °C for over 3000 cycles without obvious capacity decay. In comparison, the low-temperature performance of Li||LFP batteries is significantly degraded at -20 °C, irrespective of the use of commercial carbonate-based electrolytes (Fig. 6b) or prevalent ether-based electrolytes (Supplementary Fig. 29). This can potentially extend the application range of LMBs in low-temperature environments. This discussion has been added on **Page 21** of the Revised Supplementary Information.

3) This AILMB also demonstrates outstanding safety characteristic. The DFEA-based electrolyte delivers nonflammable properties with a SET of 0 s (inset of **Fig. 1g**), meanwhile the charged graphite cathode with DFEA-based electrolyte exhibits an endothermic peak during the DSC test (Fig. 6e). Due to the high thermostability of both electrolyte, C_n(PF₆) cathode and electrode|electrolyte interphase, the AILMB exhibits superb overcharging resistance up to 9 V without obvious change in current density, as evidenced by the low skin temperature less than 35 °C (lower-panel insets of Fig. 6f). This discussion has been added on **Page 21** of the Revised Manuscript.

Above advantages position the AILMB as a promising battery system for addressing some of the pivotal challenges currently faced by LMBs (*e.g.*, fast-charging, low-temperature limitations). Future research endeavors should be committed to bridging the energy density gap and enhancing the overall performance of AILMBs.

Moreover, as suggested by the Reviewer, we have tested the performance of LMBs with ether-based electrolytes for comparison. The corresponding description has been added into Revised Supplementary Information and the Revised Manuscript as follows:

Page 21 of the Revised Supplementary Information: “Here, the low-temperature performance of Li||LFP cells employing the widely-used Lithium bis(fluorosulfonyl) imide (LiFSI)-dimethyl ether (DME) and LiFSI-DME/1,1,2,2-tetrafluoroethyl-2,2,3,3-tetrafluoropropylether (TTE, 1: 1 by volume) electrolytes was assessed at $-20\text{ }^{\circ}\text{C}$. While the reversible capacities of Li||LFP cells can attain $\sim 82\text{ mAh g}^{-1}$ (using LiFSI-DME, **Supplementary Fig. 29a**) and $\sim 90\text{ mAh g}^{-1}$ (using LiFSI-DME/TTE, **Supplementary Fig. 29b**) at a low current density of 20 mA g^{-1} , they suffer from rapid decline at current densities higher than 100 mA g^{-1} . When charging at 100 mA g^{-1} (using CCCV mode) followed by discharging at 500 mA g^{-1} , cells based on these two electrolytes deliver similar reversible capacities of around 40 mAh g^{-1} only, although the TTE co-solvent improves the cell cyclability (**Supplementary Fig. 29c**). Furthermore, as shown in **Supplementary Fig. 29d**, the Li||LFP cells exhibit specific capacity as low as only $\sim 22\text{ mAh g}^{-1}$ (using LiFSI-DME) and $\sim 29\text{ mAh g}^{-1}$ (using LiFSI-DME/TTE) at high discharge current density of 1 A g^{-1} ”.

Page 19 of the Revised Manuscript: “It is noted that even changing the conventional carbonate-based electrolyte to ether-based electrolytes (**Supplementary Fig. 29**), the Li||LFP LMB still exhibits much lower reversible capacity in comparison with AILMB using DFEA-based electrolyte at $-20\text{ }^{\circ}\text{C}$. Above findings indicate that the fast migration/reaction kinetics of both cations and anions in AILMBs is pivotal in enhancing low-temperature applicability of LMBs”.

Supplementary Figure 29. a Typical charge-discharge profiles of the LFP-LMB using a LiFSI-DME and b LiFSI-DME/TTE electrolytes at RT and -20 °C. Current density: 20 mA g⁻¹. c Long-term cycling performance and d fast-discharging capability of the LFP-LMB at -20 °C. Employing CCCV mode for charging process at 100 mA g⁻¹, followed by discharging at various current densities.

2. Over 10000 cycles with the lithium metal anode are highly promising result. However, the evaluation condition highly alters the demonstrated cycle performances from the LMBs. For instance, the loading level of cathode, NP ratio, and the injected electrolyte amount are significantly influencing the cycle performances. Please verify these conditions. Because the thickness of Li electrode in this study was 450 μm, the Li reservoir is highly included in the cell. Since the thick lithium metal was comprised in this work, the capacity retention is highly extended even with the Li-consumption. Thus, the cell evaluation should be conducted at NP ratio below 2 for high specific energy system.

3. The electrolyte-to-capacity (E/C) ratio of the cell evaluation condition is too high. The 60 $\mu\text{L}/\text{mg}$ is flooded condition, indicating that the specific energy of LMB was below 300Wh/kg. Compared to specific energy of state-of-art lithium-ion batteries (>300Wh/kg), this evaluation condition is not meaningful for the advanced batteries system. (*Joule*, 2019, 3, 1094) The capacity retention from DFEA electrolyte should be reported based on the lean electrolyte condition ($E/C < 2.5$).

Reply: Thanks for your professional comments. We would like to reply the above two comments together as follows.

As commented by the Reviewer, critical parameters (e.g., the electrolyte-to-capacity (E/C) ratio and the capacity ratio between the anode and cathode (N/P ratio) in cell evaluation should be reduced for practical considerations¹. According, we replaced the 450 μm Li metal anode with thin Li foils, reducing the N/P ratio to ~ 5.7 (with 50 μm -thick Li foil) and ~ 2.0 (with 20 μm -thick Li foil), respectively (Supplementary Fig. 31). As for the E/C ratio, it is noted that, the slat in electrolyte is considered as active material in AILMBs, thus the amount of electrolyte is inevitably higher than that in conventional LMBs. Considering the E/C ratio is proportional to the electrolyte density (ρ_E) and inversely proportional to the concentration of the electrolyte, the salt concentration has been increased to 3.5 M to minimize the electrolyte weight. A surplus of 20% electrolyte was used to account for irreversible losses (e.g., SEI and CEI formation) and to guarantee sufficient ionic conductivity at different SOC². Accordingly, a E/C ratio of 2.75 (calculated based on the fully charged state) was used for cell performance assessment. After three activation cycles at 20 mA g^{-1} , the cells were cycled using the CCCV model for charging process (100 mA g^{-1}), followed by discharging at 100 mA g^{-1} . It is seen that the AILMB exhibits a high capacity retention of $\sim 93.0\%$ with minimal cell polarization throughout 500 cycles under N/P ratio of ~ 5.7 (Supplementary Fig. 31a), and it can sustains 140 cycles with $\sim 90.7\%$ retainable capacity under N/P ratio of ~ 2.0 (Supplementary Fig. 31b).

We have cited the significant research work (Chen et al., *Joule* 3, 1094-1105) on cell evaluation as **ref. 33** in the Revised Manuscript, have added above discussions on **Page 23** of the Revised Supplementary Information and **Page 20** of the Revised Manuscript.

Supplementary Figure 31. Cycling performance of AILMB using DFEA-based electrolyte with a N/P ratio of **a** ~ 5.7 and **b** ~ 2.0 at 25 °C, respectively. Employing CCCV mode for charging process at 100 mA g^{-1} , followed by discharging at 100 mA g^{-1} .

References:

1. Chen, S. et al. Critical parameters for evaluating coin cells and pouch cells of rechargeable Li-metal batteries. *Joule* **3**, 1094-1105 (2019).
2. Placke, T. et al. Perspective on performance, cost, and technical challenges for practical dual-ion batteries. *Joule* **2**, 2528-2550 (2018).

4. The electrolyte for high energy batteries should be evaluated at high areal capacity, and hence the symmetric cell evaluation should be performed over 3.0 mAh/cm^2 .

Reply: Following the Reviewer's suggestion, we evaluated the electrolyte performance in symmetric cells at high areal capacities of 3 mAh cm^{-2} and 4 mAh cm^{-2} (Supplementary Fig. 19), and compared our results with previous reports (Supplementary Table 3). Accordingly, we have added the discussions on Page 14 of the Revised Manuscript: "Moreover, upon further increasing the plating/stripping areal capacities, the Li||Li cells employing DFEA-based electrolyte still demonstrate

significant improvements in the cycling capability with areal capacities of 3 mAh cm^{-2} (930 h and 300 h at 1 mA cm^{-2} and 3 mA cm^{-2} , respectively) and 4 mAh cm^{-2} (730 h and 500 h at 1 mA cm^{-2} and 2 mA cm^{-2} , respectively; **Supplementary Fig. 19**). These results represent a compelling advancement comparable with reported electrolytes (**Supplementary Table 3**)”.

Supplementary Figure 19. Voltage profiles of Li||Li symmetric cells using EC/DEC and DFEA-based electrolytes with a cutoff capacity of 3 mAh cm^{-2} at **a** 1 mA cm^{-2} , **b** 3 mA cm^{-2} , and with an areal capacity of 4 mAh cm^{-2} at **c** 1 mA cm^{-2} , **d** 2 mA cm^{-2} .

Supplementary Table 3. Comparisons of the cycling stability of Li||Li cells with reported electrolytes.

Electrolyte	Areal capacity (mAh g^{-1})	Current density (mA g^{-1})	Cycle time (h)	Ref.
2 M LiFSI-TFDMP	1	1	1600	Zhao, Y. et al. Nat. Commun. 14 , 299 (2023).
0.15 M LiFSI+0.15 M LiTFSI+0.15 M LiNO ₃ -DME	1	1	800	Wang, Q. et al. Nat. Commun. 14 , 440 (2023).
1 m LiFSI-DMTMSA	1.5	0.5	1200	Xue, W. et al. Nat. Energy 6 , 495–505 (2021)
2 M LiFSI-BFE	1	1	400	Zhang, G. et al. Nat. Commun. 14 , 1081 (2023).
1 M LiPF ₆ -EC/DEC+0.1 M LiTFA+0.1 M LiNO ₃	3	3	165	Xia, Y. et al. Nat. Energy 8 , 934-945 (2023).
1 M LiPF ₆ -EC/DMC-	2	0.5	330	Li, F. et al. Angew.

BFA				Chem. Int. Ed. 60 , 6600 (2021).
0.1 M LiFSI/0.1 M LiTFSI/0.1 M LiDFOB/0.1 M LiNO ₃ /1.0 m LiPF ₆ - EC/DMC, with 5% FEC	0.5	0.5	800	Wang, Q. et al. Adv. Mater. 35 , 2210677 (2023).
1.2 M LiPF ₆ - DFEA+10%FEC	0.5	0.5	3000	This work
1.2 M LiPF ₆ - DFEA+10%FEC	1	1	1600	This work
1.2 M LiPF ₆ - DFEA+10%FEC	3	1	930	This work
1.2 M LiPF ₆ - DFEA+10%FEC	4	2	500	This work

5. With DFEA electrolyte, what is the specific energy demonstrated? The obtained specific energy should be stated. This concerns is highly correlated with flooded evaluation condition.

Reply: Thanks for the constructive comment. Accordingly, a 440 mAh Li|graphite multi-layer pouch cell was packaged to determine the specific energy of the AILMB (Supplementary Fig. 33a). The cathode with a high areal capacity of 1.64 mAh cm⁻² were incorporated to pair with 20 μm-thick Li foil deposited on Cu foils. This configuration gives a low capacity ratio between the anode and cathode (N/P ratio) of ~2.4. As discussed above, to increase the specific energy, the electrolyte amount has to be limited. Here, the electrolyte concentration has been increased to 3.5 M. Besides, an additional ~10% excess electrolyte was used to ensure sufficient ionic conductivity and to account for irreversible losses during the battery cycling. Electrolyte with a salt concentration of 3.5 M was applied considering the electrolyte amount in AILMBs is inevitably higher than that in conventional LMBs. The pouch cell was charged at 20 mA g⁻¹ using the CCCV mode, followed by discharging at a constant current rate of 20 mA g⁻¹ (Supplementary Fig. 33b). Therefore, the energy density of the AILMB is^{1,2}:

$$E_{\text{full}} = C_{\text{full}} \times V_{\text{cell}} / (m_{\text{total}})$$

where V_{cell} and C_{full} are the average working voltage and the reversible capacity of the full cell, respectively, m_{total} is the total weight based on the sum of current collector, cathode, anode, separator and electrolyte. The weight of the packing cell bag is excluded from the specific energy calculation due to our limited size of cell^{3,4}. As shown in Supplementary Table 6, the specific energy of the AILMB using DFEA-based electrolyte is calculated as ~141.7 Wh kg⁻¹. Moving forward, it is imperative to conduct further research on modifying graphite cathodes (e.g., surface treatment,

doping), optimizing electrolytes (e.g., novel anions with smaller sizes, multivalent anions with more charge numbers, and solvent with lower density), and refining engineering issues, to improve the specific energy of AILMBs without compromising their superior fast-cycling, low-temperature, and safety characteristics. We have added above result on **Page 21** of the Revised Manuscript and **Page 29** of the Revised Supplementary Information.

Supplementary Figure 33. a Optic image and **b** charge-discharge cycling of the packaged AILMB at 20 mA g^{-1} , $25 \text{ }^\circ\text{C}$.

Supplementary Table 6. Calculated specific energy of the packaged AILMB.

Component	Value
Graphite cathode (96% active material)	5.28 g
Al foil	0.53 g
Li/Cu anode	1.23 g
Electrolyte + Separator	7.14 g
Total weight (exclude packing cell bag), m_{total}	14.18
Reversible capacity, C_{full}	440.5 mAh
Average working voltage, V_{cell}	4.56 V
Specific energy, E_{full}	141.7 Wh kg^{-1}

References:

1. Betz, J. et al. Theoretical versus practical energy: a plea for more transparency in the energy calculation of different rechargeable battery systems. *Adv. Energy Mater.* **9**, 1803170 (2019).
2. Placke, T. et al. Perspective on performance, cost, and technical challenges for practical dual-ion batteries. *Joule* **2**, 2528-2550 (2018).

3. Xia, Y. et al. Designing an asymmetric ether-like lithium salt to enable fast-cycling high-energy lithium metal batteries. *Nat. Energy* **8**, 934-945 (2023).
4. Lu, Y. et al. Tuning the Li⁺ solvation structure by a “bulky coordinating” strategy enables nonflammable electrolyte for ultrahigh voltage lithium metal batteries. *ACS Nano* **17**, 9586-9599 (2023).

Reviewer #2 (Remarks to the Author):

This paper reports a new electrolyte solvent for anion-intercalation lithium batteries. The authors demonstrate that 2,2-difluoroethyl acetate (DFEA) works as a good electrolyte solvent that enables highly stable cycling of the battery. It is also shown that the difluoro part is better than usual trifluoro moiety. The solvent is novel and the battery performance is impressive. However, there are some problems that must be addressed before accepted for publication.

1. The title should include some description of “anion-intercalation” because the DFEA solvent mainly contributes to the anion-intercalation reaction. There are many solvents that show better Coulombic efficiency for Li metal anode side.

Reply: Thanks for the Reviewer’s positive comments on the quality of this work. Accordingly, the description “anion-intercalation” has been included to the title as “Difluoride ester solvent toward fast-rate anion-intercalation lithium metal batteries under extreme working conditions”.

2. For nomenclature, the authors refer this class of solvent (EA, TFEA, and DFEA) as carboxylates, but they are usually referred to as esters. Carboxylates are an anion with a structure of RCOO⁻.

Reply: Thanks for the value comment. As suggested by the Reviewer, the term “carboxylate” has been corrected to “ester” throughout the Revised Manuscript.

3. The solvation structures of Li⁺ and EA, TFEA, and DFEA should be shown. Does the partial dipole in the difluoro moiety in DFEA contribute to the Li⁺ solvation?

Reply: As suggested by the Reviewer, the solvation structures of Li⁺ and EA, TFEA, and DFEA have been investigated. The binding configurations and corresponding binding energies (ΔG) between Li⁺ and each solvent molecule were shown in Supplementary Fig. 12a-c. It is seen that Li⁺ ions form a chelating seven-member ring geometry for DFEA or TFEA whose F atoms also participate in the coordination structure. Moreover, Li⁺ exhibits stronger interaction with -CHF₂ than -CF₃, as evidenced from the shorter Li-F distance of 1.906 versus 1.954 Å for -CF₃. Such a

stronger interaction between Li^+ and the $-\text{CHF}_2$ can be attributed to the local dipole which is more negatively charged than $-\text{CF}_3$ (Supplementary Fig. 11).

To further investigate the Li^+ solvation sheath, the RDFs and coordination numbers were obtained from MD simulations (Supplementary Fig. 12d-h). It is worth noting that the coordination of Li^+ to F atoms can be seen for DFEA and TFEA electrolytes. Clearly, the $\text{Li}-\text{F}$ RDFs displays that more F atoms on $-\text{CHF}_2$ participating in Li^+ solvation compared with that on $-\text{CF}_3$. These results are consistent with the charge distribution and binding energy in Supplementary Fig. 11 and Supplementary Fig. 12a-c. The above discussions have been added on Page 8 of the Revised Supplementary Information. Furthermore, the description “In addition, the DFEA- Li^+ interaction is enhanced due to the existence of local dipole on the $-\text{CHF}_2$ (Supplementary Fig. 12d-h), which is similar to the recent report by Bao et al.” has been included on Page 11 of the Revised Manuscript.

Supplementary Figure 12. Coordination structures and binding energies of a Li^+ -EA, b Li^+ -DFEA and c Li^+ -TFEA. Li^+ RDF obtained from MD simulations of d EA, e, g DFEA and f, h TFEA electrolytes. Solid lines represent $g(r)$ while dashed lines represent coordination number.

References:

1. Yu, Z. et al. Rational solvent molecule tuning for high-performance lithium metal battery electrolytes. *Nat. Energy* 7, 94-106 (2022).

4. *The comparison in Fig. 1 is not fair because FEC is added to only DFEA. This fact is not described in the Figure or the caption. The authors claim that the amount of FEC is small but the addition of 10 wt% is very large. Usually, additives are used less than 5 wt%. This figure misleads the readers, so it should be revised.*

Reply: Thanks for the Reviewer's valuable comment. As suggested by the Reviewer, 10 wt% FEC addition has been added to the Figure caption (Page # of the Revised Manuscript): "**Fig. 1 | h DFEA (with 10 wt% FEC) electrolytes. i DFEA (with 10 wt% FEC) electrolytes.**" Moreover, we have revised the statement of "FEC additive" on Page 6 of the Revised Manuscript: "**The commonly used fluoroethylene carbonate (FEC, 10 wt%) was added to the 1.2 M LiPF₆ salt in DFEA electrolyte (Supplementary Fig. 5), henceforth referred to as DFEA-based electrolyte**". The "sole-solvent" to describe DFEA has been accordingly changed to "solvent" throughout the Revised Manuscript.

5. *This FEC addition causes the same problems in Figs. 4-6. It is well known that FEC works as a very good additive to enable highly reversible Li plating/stripping. So, it is not clear whether the high cycling performance is derived from DFEA or FEC. Furthermore, since FEC forms LiF-rich thin SEI on Li metal, the observed SEI components in Fig. 5 may primarily be derived from FEC, not from DFEA, which significantly compromises the impact and novelty of this work.*

Reply: Thanks for the valuable comments. We would like to respond to this comment from the following aspects:

1) We agree with the Reviewer that FEC acts as a useful electrolyte additive for highly reversible Li plating/stripping. Following the Reviewer's suggestion, the cycling behavior of Li||Li cells based on the DFEA (without FEC) and the EA (with 10 wt% FEC) electrolytes were investigated at 0.5 mA cm⁻². It is seen that the EA electrolyte with FEC addition exhibits a large overpotential and quickly suffers from short-circuiting (Supplementary Fig. 23a). In contrast, the cell using DFEA (without FEC) electrolyte shows a smaller overpotential and longer lifespan (Supplementary Fig. 23b). In addition, the FEC addition into the DFEA electrolyte can enable highly reversible Li plating/stripping (Fig. 4a). These results suggest that it is the combined effect of FEC and DFEA that endows the Li||Li cell with long-term cycling stability. Besides, it is seen that AILMBs using EA with 10 wt% FEC electrolyte fails to support the anion-intercalation chemistry on the graphite cathode (Supplementary Fig. 23c). In contrast, the DFEA (both without and with FEC addition) facilitates a reversible anion de-/intercalation process on the graphite host without solvent co-intercalation (Fig. 1, Supplementary Fig. 5 and Supplementary Fig. 23d). These results indicate

that it is the DFEA solvent, rather than the FEC addition, that supports the reversible cathode reaction. The above discussion has been added on Page 15 of the Revised Supplementary Information and Page 17 of the Revised Manuscript.

Supplementary Figure 23. Voltage profiles of Li||Li cells using **a** EA (10 wt% FEC) and **b** DFEA (without FEC) electrolyte at 0.5 mA cm^{-2} with a cutoff capacity of 0.5 mAh cm^{-2} . **c** Voltage curves of Li||graphite cells using the EA (10 wt% FEC) electrolyte at 20 mA g^{-1} at 25°C . **d** Cycling performance of Li||graphite cells using DFEA (without FEC) electrolyte at 1 A g^{-1} after three activation cycles at 20 mA g^{-1} at 25°C .

2) Moreover, as commented by the Reviewer, it is known that FEC can form LiF-rich thin SEI on Li metal. In accordance with the Reviewer's suggestion, the SEI components were characterized by in-depth XPS and TEM after cycling the Li||Li cell in DFEA (without FEC) and DFEA-based electrolytes. As displayed in **Supplementary Fig. 24** and **Supplementary Fig. 25**, a large amount of LiF component is observed throughout the whole sputtering process for both electrolytes, indicating that both DFEA (without FEC) and DFEA-based electrolyte contribute to the formation of a LiF-rich SEI layer on the Li metal. It is noted that the FEC addition significantly inhibits the decomposition of both LiPF_6 salt and solvents, as evidenced from the larger amounts of $\text{Li}_x\text{PO}_y\text{F}_z$ and organic species (e.g., ROCO_2Li) derived from the DFEA (without FEC) electrolyte. In addition, TEM was conducted to investigate the microstructures of SEI shells onto a Cu grid from Li||Cu cells cycled in DFEA (without FEC) electrolyte. As displayed in **Supplementary Fig. 25**, the lattice spacing of 2.03 \AA corresponds to the (200) crystal plane of LiF nanoparticles, indicating that both DFEA (without FEC) and DFEA-based (**Fig. 5b**) electrolytes lead to the LiF formation on Li anode. However, it is noted that despite the formation of a LiF-enrich SEI with the DFEA (without FEC) electrolyte, the undesired decomposition of the DFEA solvent is not effectively suppressed, as seen from the presence of Li_2O ((111) plane, 2.70 \AA) nanoparticles

(Supplementary Fig. 25). These findings are consistent with the XPS results, further proving that both FEC and DFEA solvent contribute to the LiF-rich SEI on the Li metal, while the FEC addition effectively suppress the excessive decomposition of DFEA solvent. The above discussion has been added on **Page 17** of the Revised Supplementary Information and **Page 17** of the Revised Manuscript.

Supplementary Figure 24. F 1s XPS depth profiles of the Li metal after using **a** DFEA (without FEC) and **b** DFEA-based electrolytes. LiF: 685 eV; $\text{Li}_x\text{PO}_y\text{F}_z$: 687 eV; CF_3 : 688.5 eV. C 1s XPS depth profiles of the Li metal using **c** DFEA (without FEC) and **d** DFEA-based electrolytes. C-C/C-H: 284.5 eV, C-O: 285.5 eV, C=O: 286.7 eV, CO_3^{2-} : 288.8 eV, CF_3 : 289.8 eV.

Supplementary Figure 25. TEM image of the SEI shell formed by plating/stripping Li on a Cu grid using DFEA (without FEC) electrolyte.

REVIEWERS' COMMENTS

Reviewer #1 (Remarks to the Author):

The authors adequately address the concerns raised by reviewer. Hence, reviewer recommends accept of the manuscript.

Reviewer #2 (Remarks to the Author):

The questions have been well addressed, and the manuscript has been well improved. The effect of difluorosolvent becomes clear. This reviewer recommends its publication.

Comment

For the title, "difluoroester" may be more appropriate than "difluoride ester".

Response to Reviewers' Comments

Reviewer #1 (Remarks to the Author):

The authors adequately address the concerns raised by reviewer. Hence, reviewer recommends accept of the manuscript.

Reply: Thanks for the Reviewer' positive comment of this work.

Reviewer #2 (Remarks to the Author):

The questions have been well addressed, and the manuscript has been well improved. The effect of difluorosolvent becomes clear. This reviewer recommends its publication.

Comment

For the title, "difluoroester" may be more appropriate than "difluoride ester".

Reply: Thanks for the Reviewer's positive comment on the quality of this work. Accordingly, the title has been revised as "Difluoroester solvent toward fast-rate anion-intercalation lithium metal batteries under extreme conditions".